# Micro-occurrence characteristics and charging mechanism in continental shale oil from Lucaogou Formation in the Jimsar Sag, Junggar Basin, NW China

Jiasi Li, Aimin Jin [ORCID]*[☯], Rong Zhu*[☯], Zhanghua Lou, on behalf of The Hebei Scolike Petroleum Technology Co., Ltd[¶]

Institute of Marine Geology & Resources, Ocean College, Zhejiang University, Zhoushan, Zhejiang, China

☯ These authors contributed equally to this work.
¶ Membership of the Hebei Scolike Petroleum Technology Co., Ltd is provided in the Acknowledgments.
* aiminjin@163.com (AJ); zhurong@zju.edu.cn (RZ)

**Data Availability Statement:** All relevant data are within the paper and Supporting Information files.

## Abstract

The micro-occurrence characterization of shale oil is a key geological issue that restricts the effective development of continental shale oil in China. In order to make up for the lack of research in this area, this paper carries out a series of experiments on the shale oil of the Lucaogou Formation using a multi-step extraction method, with the aim of exploring the micro-occurrence types and mechanisms of shale oil in the Lucaogou Formation, as well as exploring its direct connection with production and development. In this paper, shale oil in the reservoir is divided into two categories: free oil and residual oil. The polar substances and OSN compounds are the key factors determining the occurrence state of shale oil. Abundant polar substances and OSN compounds can preferentially react with mineral surfaces (including coordination, complexation, ionic exchange, and so on) to form a stable adsorption layer, making it difficult to extract residual oil in actual exploitation. Free oil is mainly composed of aliphatic hydrocarbons, and its adsorption capacity is related to the length of the carbon chain, i.e. long carbon chain, strong adsorption capacity, and poor movability. Free oil is widely stored in pores and cracks, and that with high mobility can be the most easily extracted, making it the main target at present exploitation. In the current state of drilling and fracturing technology, research should prioritize understanding the adsorption and desorption mechanisms of crude oil, particularly residual oil. This will help optimize exploitation programs, such as carbon dioxide fracturing and displacement, to enhance shale oil production.

## 1. Introduction

The micro-occurrence of shale oil refers to the form in which shale oil exists within the pore spaces. Highly developed micro-nano pore-throat systems and extreme heterogeneity within shale formations, leading to a widely dispersed and intricate distribution of shale oil. This results in a diverse range of shale oil occurrence states, posing significant research challenges

**Funding:** This research was financially supported by the National Science and Technology major projects (No. 2011ZX05002-006-003HZ). The funder for this study, Professor Zhanghua Lou, provided financial support for the research and was actively involved in the manuscript's development and revision. The funders had no role in study design, data collection and analysis, decision to publish, or preparation of the manuscript.

**Competing interests:** The authors have declared that no competing interests exist.

[1, 2]. How dose crude oil occurrence in the microscopic pores, and what is the mechanisms governing these occurrence states? This is a key geological issue challenging the development of continental shale oil in China and serves as the theoretical foundation for determining whether shale oil formations can be effectively utilized and extracted [3].

Previous research on the Lucaogou Formation shale oil reservoir has focused on the characteristics of its source rocks, reservoir properties, and the stages of hydrocarbon accumulation. These studies have confirmed that the Lucaogou Formation possesses high-quality source rocks, and the crude oil is generally self-generated and self-stored. The micro-nano pores within the reservoirs are typically oil-bearing, exhibiting high oil saturation and superior resource potential [4]. Nonetheless, the actual extraction results have fallen short of expectations. Variances in production efficiency among distinct horizontal wells are apparent, characterized by a notable post-volume fracturing production decline (approximately 60% after the first year), substantial energy dissipation, and a limited primary recovery rate. Achieving effective development remains a significant challenge. In cases where geological and engineering conditions are similar, the significant variations in production among different wells are not well understood. Some researchers have attempted to explain this problem by examining the occurrence states of shale oil. Under scanning electron microscopy, they have observed the presence of abundant oil films adsorbed onto mineral surfaces and have noted that these oil films exhibit limited mobility [5]. Some scholars have used pyrolysis techniques to obtain the parameter $S_1$, which represents the content of free hydrocarbons, confirming the presence of free oil [6–8]. Furthermore, scholars have utilized molecular dynamics simulations, high-speed centrifugation, and nuclear magnetic resonance experiments to investigate the mobility of hydrocarbons in various occurrence states. And explored the factors influencing hydrocarbon mobility from a molecular perspective [9, 10]. Typically, free oil exhibits the highest mobility and is the primary target for extraction at present [10]. However, in the extraction of continental shale oil in China, horizontal wells subjected to extensive fracturing often exhibit characteristics of short self-spray periods and rapid production declines. Even during the transition to the pump swabbing stage, achieving stable production remains challenging, with a significant amount of residual shale oil [11]. Some researchers have found that measures such as microwave heating can enhance production [9, 12, 13]. This improvement may be attributed to the activation of some of the less mobile adsorbed crude oil. Therefore, the micro-occurrence states of shale oil may be a crucial factor contributing to the suboptimal performance of continental shale oil extraction in China and the significant variations in production among horizontal wells. Thus, it is imperative to conduct an in-depth investigation into the micro-occurrence characteristics and mechanisms of shale oil in the Lucaogou Formation.

Solvent extraction is a common method for transferring compounds from solids or solutions into a solvent. It is widely applied in evaluating recoverable oil resources and reconstructing the organic matter (OM) filling history [14–31] (Table 1). In this study, we employed a multi-step sequential extraction technique and conducted a detailed analysis of the extracted shale oil fractions using group component analysis, gas chromatography-mass spectrometry (GC-MS), infrared spectroscopy, and organic elemental detection. Additionally, 2D nuclear magnetic resonance (2D-NMR), scanning electron microscopy (SEM), and laser confocal SEM (CLSM) tests were conducted on core samples before and after extraction. The purpose is to address the inadequacies in research concerning the micro-occurrence characteristics of shale oil in the Lucaogou Formation in the Jimsar Sag. The main achievements of this study are as follows: 1. Defining micro-occurrence types of shale oil with practical production significance. 2. Providing initial explanations for the micro-occurrence characteristics and mechanisms of shale oil. 3. Developing a model for the occurrence of shale oil in micro-nano pores and reconstructing the filling history of shale oil.

**Table 1. A list of common solvent(s) used in core cleaning/extraction in literature.**

| Organic solvent | Polarity | Particle size | Temperature | Extracted shale oil | Reference |
|---|---|---|---|---|---|
| $CH_2Cl_2$ | 3.4 | Core plug | - | Free oil | [19] |
| | | 80–100 mesh | Ambient temp | Adsorbed oil | [20] |
| $CHCl_3$ | 4.4 | 100 mesh | - | Adsorbed oil | [21] |
| $CH_3OH$ / $C_3H_6O$ / $CH_2Cl_2$ (1:1:1.5, v/v) | 6.6/5.4/3.4 | <60 mesh | - | Free oil | [22] |
| $CS_2$ / $C_5H_9NO_3$ (1:1, v/v) | 2.64/11.3 | <60 mesh | - | Adsorbed oil | [22] |
| $CH_2Cl_2$ / $CH_3OH$ (93:7, v/v) | 3.4/6.6 | 0.1–0.25 mm | - | Free oil / Adsorbed oil / Inclusion oil | [23] |
| | | Clay minerals | - | Free oil / Adsorbed oil | [24] |
| | | 0.10–0.30 mm | - | Free oil / Adsorbed oil | [23] |
| | | -1 cm | Cold water | Free oil | [25] |
| | | 1 mm | | Free oil | |
| | | 150 mesh | | Adsorbed oil | |
| $C_4H_8O$ / $C_3H_6O$ / $CH_3OH$ (50:25:25, v/v) | 4.2/5.4/6.6 | Core plug | - | Adsorbed oil | [26] |
| | | 150 mesh | Cold | Adsorbed oil | [25] |
| $C_5H_5N$ | 5.3 | Kerogen | 80˚C | Bitumen | [27] |
| | | Kerogen | - | Heavy compounds | [28] |
| $CH_3OH$ | 6.6 | Kerogen | - | Bitumen | [27] |
| $C_8H_{10}$ | 2.5 | Pulverized rock | 100˚C | High molecular weight hydrocarbons | [29] |
| $C_6H_{12}$ | 0.1 | Finely-ground | 80˚C | High molecular weight hydrocarbons | [30] |
| $C_5H_{12}$ | 0 | Kerogen | - | Free oil | [28] |
| $CH_2Cl_2$ / $CH_3OH$ (9:1 v/v) | 3.4/6.6 | 8–10 mm | Room temp | Free oil | [17] |
| | | 2–5 mm | Room temp | Adsorbed oil | |
| | | 60–80 mesh | Room temp | Residual oil | |
| $CHCl_3$ / $CH_3OH$ azeotrope (1:1 v/v) | 4.4/6.6 | Core plug | - | Adsorbed oil | [19] |
| $CHCl_3$ / $C_2H_5OH$ (98:2 v/v) | 4.4/4.3 | Powdered rock | - | Hydrocarbon fluids | [31] |

## 2. Geological setting

The Jimsar Sag is located in the eastern Junggar Basin, Northwest China (Fig 1A), with an area of 1278 km$^2$. It is a dustpan-like sag developed on the folded basement of the Middle Carboniferous [32]. The sag is characterized by large sedimentary thickness in the west and gradual uplift and denudation to the east [33, 34]. It is adjacent to several sub-tectonic units in the Junggar Basin, including the Shaqi Bulge, Beisantai Bulge, and Guxi Bulge (Fig 1B). Multiple faults are developed in the sag, such as the Xidi Fault, the South No.1 Fault of Qing1 Well, the Jimsar Fault, the Fukang Fault, the Santai Fault, and the Houbaozi Fault (Fig 1B).

Oil in the Lucaogou Formation in Jimsar Sag is a typical representative of continental shale oil in China, with shale oil resources reaching 1.112 billion tons, making it a significant shale oil production demonstration area. The thickness of the Lucaogou Formation ranges from 25 to 300 m, with an average of 200 m; the burial depth is 800–4500 m, with an average of 3570 m. The lithology is complex, including dolomite, mudstone, sandstone, limestone, and so on. They are distributed in a thin interlayer structure in the stratigraphic profile [35]. The oil saturation and TOC content in the entire Lucaogou Formation shale vary significantly, fluctuating between 0–100% and 0–20% respectively (Fig 1C). The rocks contain up to 12 minerals, indicating a low maturity of the components. There is no obvious boundary between source rocks and reservoirs. Source rocks are mainly composed of dolomite (mud-crystal and micro-crystal) and mudstone, with high oil saturation. The OM types are mainly Type I and Type II. The total organic carbon (TOC) content is above 3% on average. $R_o$ is between 0.5 and 1.1 [36]. The reservoirs contain a substantial number of micro and nano-pores, as depicted in

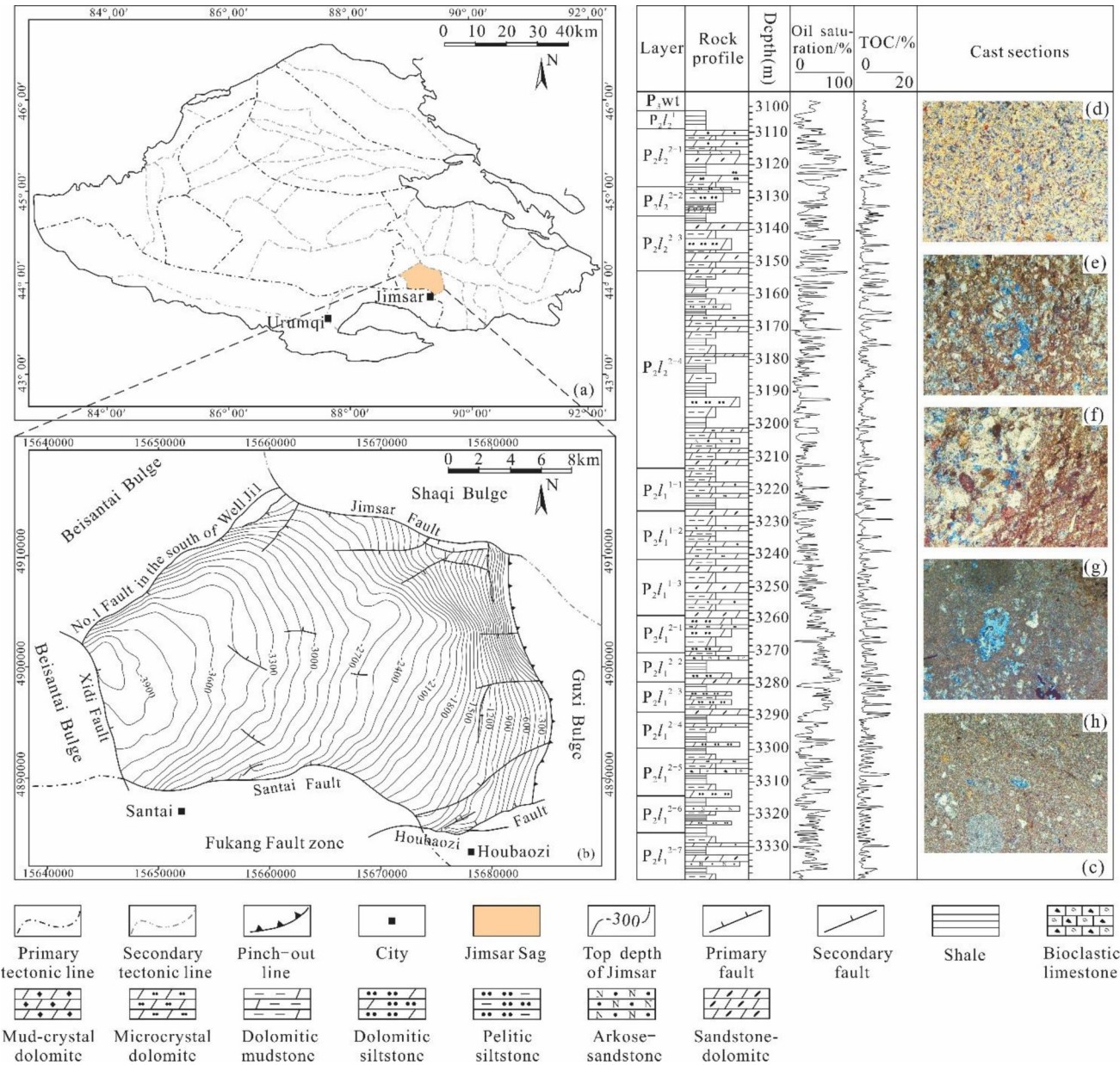

**Fig 1. Geological synthesis schematic of Jimsar Sag, Junggar Basin.** (a) Location and structural zoning of the Jimsar Sag in the Junggar Basin; (b) Structural details of the Jimsar Sag within the Junggar Basin; (c) Stratigraphic division, oil saturation and petrophysical characteristics of the Lucaogou Formation in the Jimsar Sag; (d) 3190.57 m, arkose, dissolved pores, 100×; (e) 3144.84 m, dolomitic siltstone, dissolved pores, 100×; (f) 3143.82 m, psammitic dolomite, dissolved pores, 100×; (g) 3139.70 m, mud-crystal dolomite, dissolved pores, 50×; (h) 3138.76 m, microcrystal dolomite, dissolved pores, 100×.

Fig 1D–1H, and their thickness typically does not exceed 4 meters. The shale oil's occurrence in these reservoirs is characterized by its intricate and varied states, rendering them challenging to exploit. To achieve effective production capacity, extensive volume fracturing is still necessary. However, a common issue is the rapid production decline during the self-spray stage, resulting in short stable production cycles during the pump swabbing stage. Studying the micro-occurrence characteristics of shale oil allows us to understand the interactions between oil and rock at a microscopic scale and the mobility of shale oil, which contributes to addressing this issue.

## 3. Samples and experiments

### 3.1 Sample selection and representation

The main oil-bearing strata of the Jimsar shale oil are distributed in the $P_2l_2$ and $P_2l_1$ groups, with an average thickness of 38 meters and 44 meters. These strata are characterized by saline lake facies intercalated with deltaic facies, and they exhibit a complex composition of rock and mineral types. Currently, the primary focus of oil exploitation in this region is on the reservoirs that have been filled and modified by dolomitic siltstone, psammitic dolomite, and arkose [37].

We collected two samples from the main oil-bearing strata using sealed coring. Sample A, located in the $P_2l_2^{2-1}$ layer, consists of dolomitic siltstone (Fig 2A and 2B), while Sample B, from the $P_2l_1^{2-1}$ layer, comprises psammitic dolomite (Fig 2C and 2D). These samples represent the predominant oil-bearing layers and lithologies within the Lucaogou Formation. Investigating the micro-occurrence characteristics and mechanisms of shale oil in these two core samples will contribute to a comprehensive understanding of the primary oil-bearing strata in the Jimsar shale oil region, providing valuable insights for shale oil research in this area.

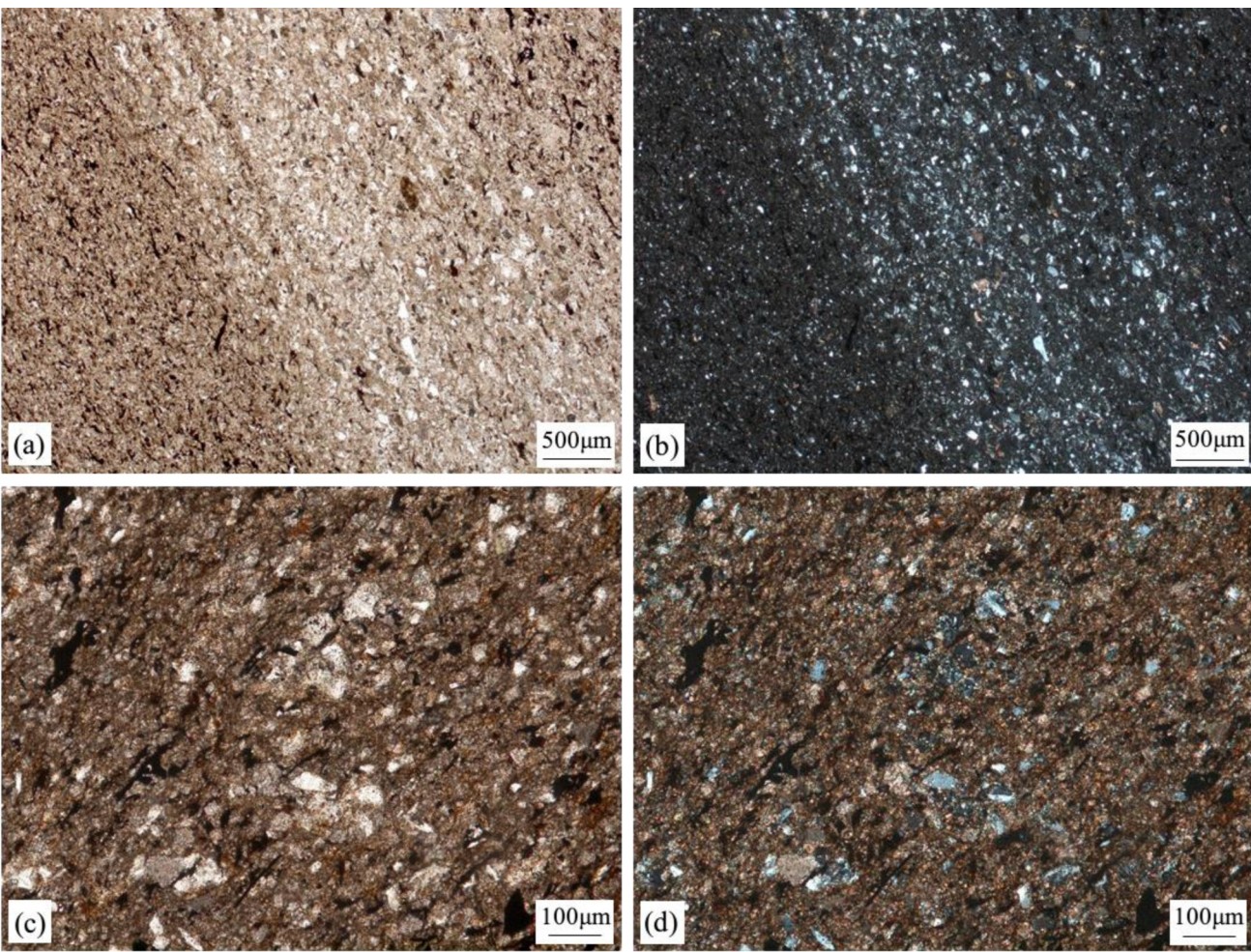

**Fig 2. Petrographic characteristics of sample rock slices.** (a) Sample A, 2677.25 m, dolomitic siltstone, single polarized light; (b) Sample A, 2677.25 m, dolomitic siltstone, orthogonal light; (c) Sample B, 3630.20 m, psammitic dolomite, single polarized light; (d) Sample B, 3630.20 m, psammitic dolomite, orthogonal light.

After the samples were removed, they were enveloped in paraffin and refrigerated to minimize the potential loss of hydrocarbons, particularly light hydrocarbons. All experiments were conducted at the Hebei Scolike Petroleum Technology Co., Ltd.

### 3.2 2D-NMR

Nuclear magnetic resonance (NMR) can non-destructively detect hydrogen nuclei signals in rock samples. We adopt 2D-NMR to simultaneously obtain the transverse and longitudinal relaxation times (i.e. $T_1$ and $T_2$) of hydrogen nuclei, effectively distinguish different categories of hydrogen-containing components, and quantitatively calculate the signal proportion of each component, i.e. fluid saturation [38–43]. The experimental equipment is MicroMR12-040V, a high-precision benchtop NMR non-conventional core analyzer that is manufactured in the United States. The NMR frequency is 20 MHz, the experimental temperature is constant at 25˚C, and the test sequence is SR-CPMG.

First, the signal collection parameters were set according to the sample conditions (i.e., recycle delay: 20 ms, number of scans: 32, echo spacing for CPMG: 0.07 ms, number of CPMG echoes: 6000, estimated $T_2$ Max: 61 ms, 90˚C-180˚C pulse separation 11.84 ms). Then, the original samples were placed into the instrument and verified to correct parameters before the NMR signals were acquired. Following this step, an inversion of the collected results was conducted using the software provided with the instrument. Finally, the components were classified and quantitatively interpreted based on the inversion results.

### 3.3 Multi-stage sequential extraction

Samples in different particle sizes were extracted using different polar solvents. Through the investigation (Table 1), we selected two combinations of solvents with high extraction efficiency that are allowed to be used in the laboratory. The weakly polar solvent group is $CH_2Cl_2$ / $CH_3OH$ (93:7, v/v), while the strongly polar solvent group is $CH_2Cl_2$ / $CH_3OH$ / $CH_3H_6O$ (50:25:25, v/v). The heating method chosen is the water bath method, with a temperature set at 50˚C. The entire extraction process is divided into four steps:

Step 1: Samples A and B were manually crushed into blocks (size about 1cm×1cm×1cm). Two blocks of the original samples were retained for other tests, and the remaining samples were placed into a Soxhlet extractor for extraction. Obtained by continuous extraction using the weakly polar solvent group until the solvent was colorless, it took 168 hours for Samples A and B.

Step 2: The samples were further manually crushed to approximately 0.1 cm, with one piece of each sample retained for additional tests. In the second step of extraction, we used the same solvent and extraction method as in step 1, continuing until the solution became colorless. This process lasted for 48 hours.

Step 3: The samples from step 2 were further manually ground to 80–120 mesh powder using a mortar. A portion of both samples was set aside for other tests. Extract III was obtained using the same method and solvent as in step 2. It took 72 hours to reach the colorless stage in the third-step extraction to get Extract III.

Step 4: After the third-step extraction, a small portion of each of the two samples was set aside for future use. The remaining samples were subjected to extraction using a strong polar solvent group and the same extraction method, continuing until the solvent turned colorless, resulting in the production of Extract IV, which took 72 hours.

### 3.4 Group components and GC-MS

The separation of group components was conducted in accordance with Chinese industry standard SY/T 5119–2016. Following this, we tested the separated saturated hydrocarbons

using the TSQ8000 Evo meteorological chromatography-mass spectrometer. Helium, with 99.999% purity and a flow rate of 1 mL/min, served as the carrier gas. The chromatography column, an HP-5MS with dimensions of 60 m×0.25 mm×0.25 μm, operated with an inlet temperature of 300˚C and no split injection. We programmed the temperature rise as follows: the initial temperature was set at 50˚C and held for 1 minute, then raised to 120˚C at a rate of 20˚C/min, and further increased to 310˚C at a rate of 3˚C/min, where it was maintained for 30 minutes. The mass spectrometry ionization method employed was EI (70 eV), and we selected the data acquisition method as ion scan (SIM).

### 3.5 Laser confocal SEM

The sample's overall characteristics were first observed using optical microscopy, and regions rich in organic matter (OM) were selected to obtain single polarized light and orthogonal light microscopic views. Subsequently, selected areas underwent fluorescence scanning to generate phase fluorescence images. Layer-by-layer scanning of the sample was performed using a laser confocal microscope to acquire the distribution characteristics of light and heavy components in each layer. A 2D map illustrating the distribution of these components was synthesized. The collected volume data were then utilized to create a 3D distribution map of OM. Different components were represented by distinct colors based on the received wavelength signal, with light components shown in blue and heavy components in red.

### 3.6 Infrared spectrum

The infrared spectroscopy test for crude oil was conducted in accordance with the Chinese industry standard, No. GBT 6040–2002. Due to limitations in sample quality and volume, the quantity of shale oil extracts available was insufficient to meet the requirements for the liquid-film method. As a result, we adopted the solution method to prepare the sample, used $CCl_4$ (less IR absorption) as a solvent to dissolve shale oil extracts, and tested it in the fluorescence infrared-chromatography coupler through the Fourier variation infrared spectroscopy detection module.

### 3.7 Organic element

The test was carried out at room temperature using a Vario-MICRO elemental analyzer. Initially, we ignited the samples within a high-temperature combustion tube supplied with oxygen. This process led to the oxidation of carbon, hydrogen, and nitrogen into carbon dioxide, water, and nitrogen oxides, respectively. Then, through the reduction tube, the nitrogen oxide was reduced to nitrogen gas. The resulting carbon dioxide, water, and nitrogen gas were detected to calculate the content of carbon, hydrogen, and nitrogen elements in the samples. The oxygen in the organic matter of the samples was cracked in the cracking tube at a high temperature. We detected the generated carbon monoxide and calculated the content of the oxygen element.

## 4. Results and discussion

### 4.1 Hydrogen-containing components in the samples

Two-dimensional (2D) NMR can detect hydrogen nuclei signals in solid OM, hydrocarbon fluids, and non-hydrocarbon fluids in a sample, which benefits distinguishing the hydrogen-containing components [39, 40]. Three components, i.e. solid OM, hydroxyl/bound water, and soluble OM, were identified in samples A and B (Fig 3). The content of each component was expressed as the ratio of the volume of the component to the mass of the sample in ul/g.

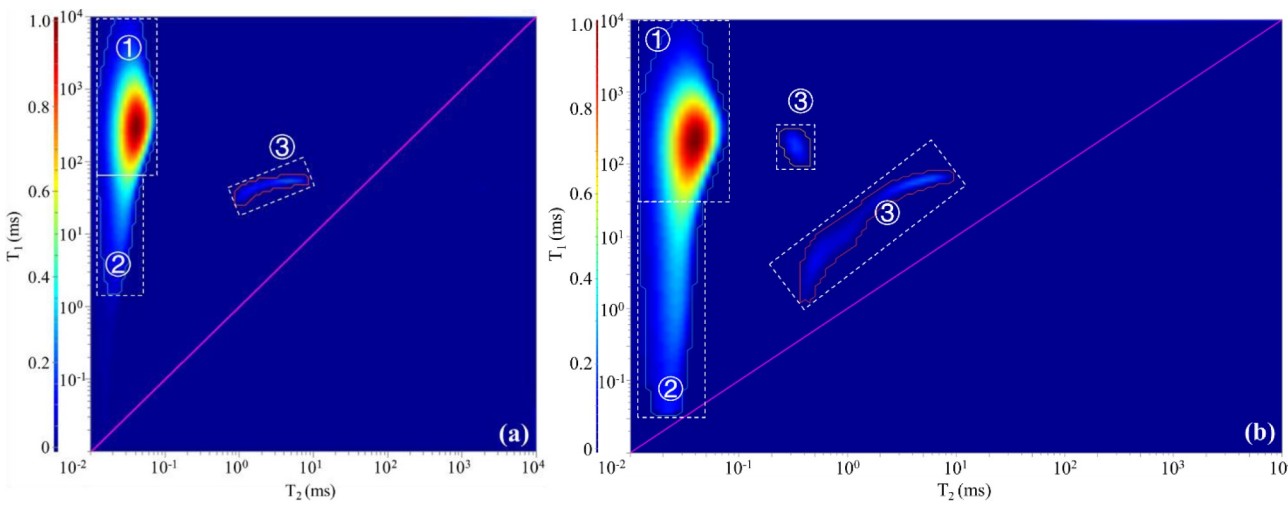

**Fig 3. Identification of hydrogen-containing components in two samples using 2D-NMR.** (a) The result of original sample of A; (b) The result of original sample of B. Type of hydrocarbon-containing component: ① Solid OM; ② Hydroxyl/bound water; ③ Soluble OM.

The volume of the hydrogen-containing components detected in sample A is 1166.42 ul, with 57.75 ul/g of hydroxyl/bound water, 217.82 ul/g of solid OM, and 4.75 ul/g of soluble OM. The volume of the hydrogen-containing components detected in sample B is 2439.95 ul, with 118.85 ul/g of hydroxyl/bound water, 226.44 ul/g of solid OM, and 9.82 ul/g of soluble OM. The ratios of solid OM to soluble OM of these two samples are 45.86 and 23.06, respectively.

According to the definition in organic geochemistry, OM in organic-rich shale formation can be divided into two types, i.e. kerogen, and bitumen. Kerogen is a dispersed OM in sedimentary rocks, which is insoluble in alkali, non-oxidizing acids, and non-polar organic solvents [43]. Bitumen is a general term used to describe various hydrocarbons with different viscosities, such as petroleum, bitumen mineral, and other non-hydrocarbon components [44–48]. In organic geochemistry, bitumen refers to soluble OM that can be extracted with organic solvents (e.g. chloroform) [49–51]. In organic petrology, bitumen usually refers to a secondary dispersed OM, including solid bitumen and pyrobitumen [52, 53]. According to the different stages of hydrocarbon generation, solid bitumen can be divided into pre-oil solid bitumen and post-oil solid bitumen [46]. The latter is the product after the peak of oil generation ($R_o$ = 0.8%~1.0%). Pyrobitumen is a product within the dry gas window, which can be converted from post-oil solid bitumen, or can be the product of secondary oil cracking [53–57]. The OM types of the Lucaogou Formation are Type I and Type II, with $R_o$ ranging between 0.7% and 1.1%, which is at the peak of oil generation [36]. SEM results show that a large amount of solid bitumen developed in both samples. Morphologically, there are two types of pores in samples: ① no pores development and smooth surface (Fig 4A and 4B); ② pores with isolated and honeycomb (Fig 4C and 4D). Considering that oil generation is a continuous process, the pre-oil solid bitumen, post-oil solid bitumen, and bitumen in oiling may coexist [55, 56]. Therefore, the non-porous solid bitumen in the samples may be pre-oil solid bitumen. Bitumen with scattered isolated pores may belong to the post-oil solid bitumen. The bitumen with honeycomb pores may be post-oil solid bitumen; alternatively, it may be residual pyrobitumen from the OM cracking and gas generation process, which are difficult to distinguish.

Although the content of soluble OM in the samples is relatively low (Fig 3), some high oil-bearing grade core samples in the study area show soluble OM under SEM. Soluble OM mainly occurs in pores and fractures and is generally adsorbed on the surface of mineral

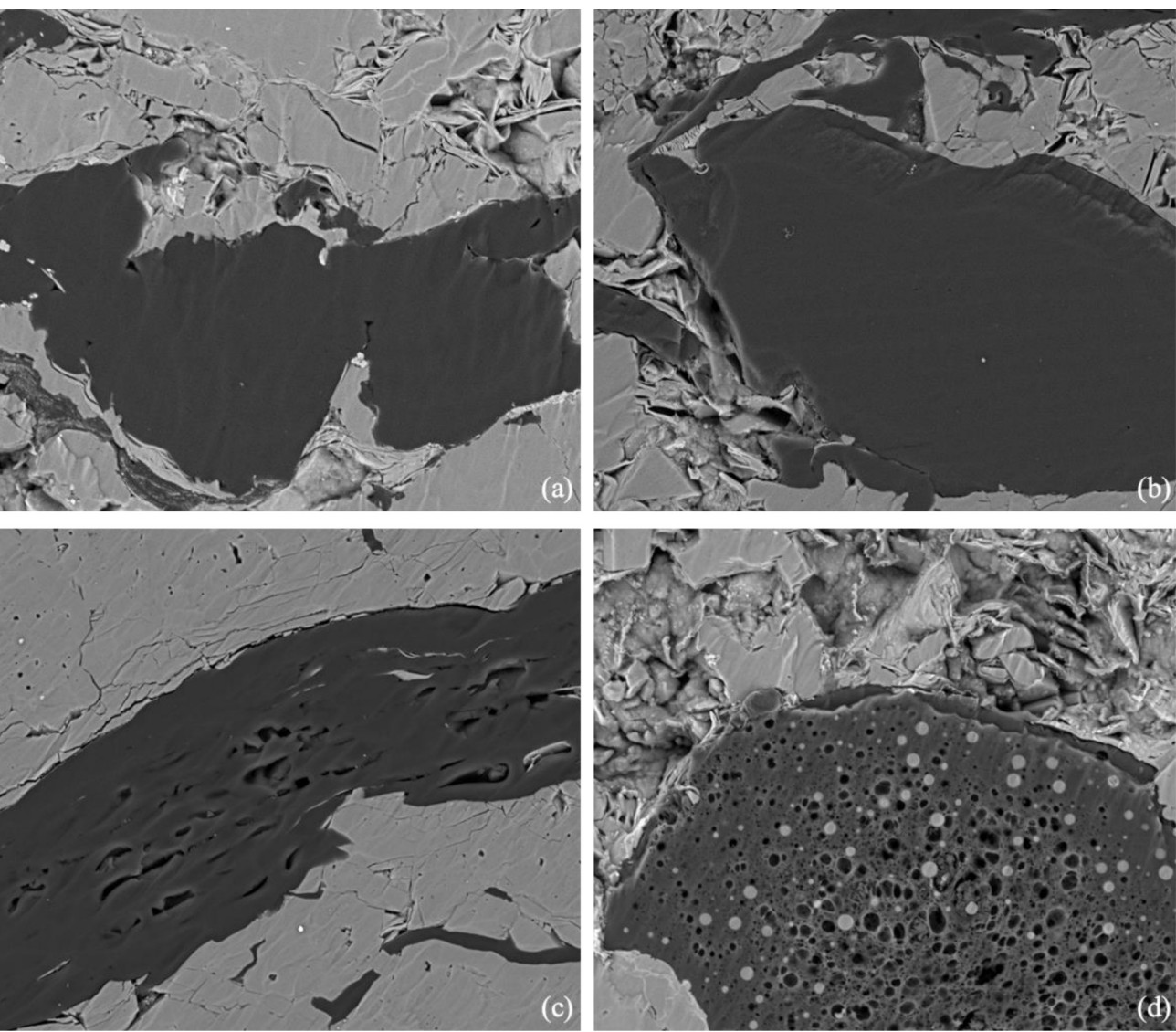

**Fig 4. Characterization of the solid OM in samples A and B after extraction.** (a) Sample A, solid OM with no pores and smooth surface; (b) Sample B, solid OM with no pores and smooth surface; (c) Sample B, solid OM with scattered isolated pores; (d) Sample A, solid OM with honeycomb pores.

particles (mainly feldspar, quartz, dolomite, and clay minerals) (Fig 5). We did not observe free state soluble OM under SEM, which is presumably related to the environmental conditions. Free oil may be liquid under original formation conditions. But after the samples were drilled from the stratum, the original temperature and pressure environment were altered. The viscosity and density of liquid shale oil increased, converting it to a solid-like state. Furthermore, during the later preservation and experiments, some light hydrocarbons were unavoidably dissipated, and the amount of loss increases with time. Therefore, it is difficult to observe the soluble OM in the free state under the microscope.

## 4.2 Hydrogen-containing components before and after extraction

The soluble OM in samples A and B is the main target of this extraction. We used 2D-NMR to detect the hydrogen-containing components in the residual samples after each extraction step

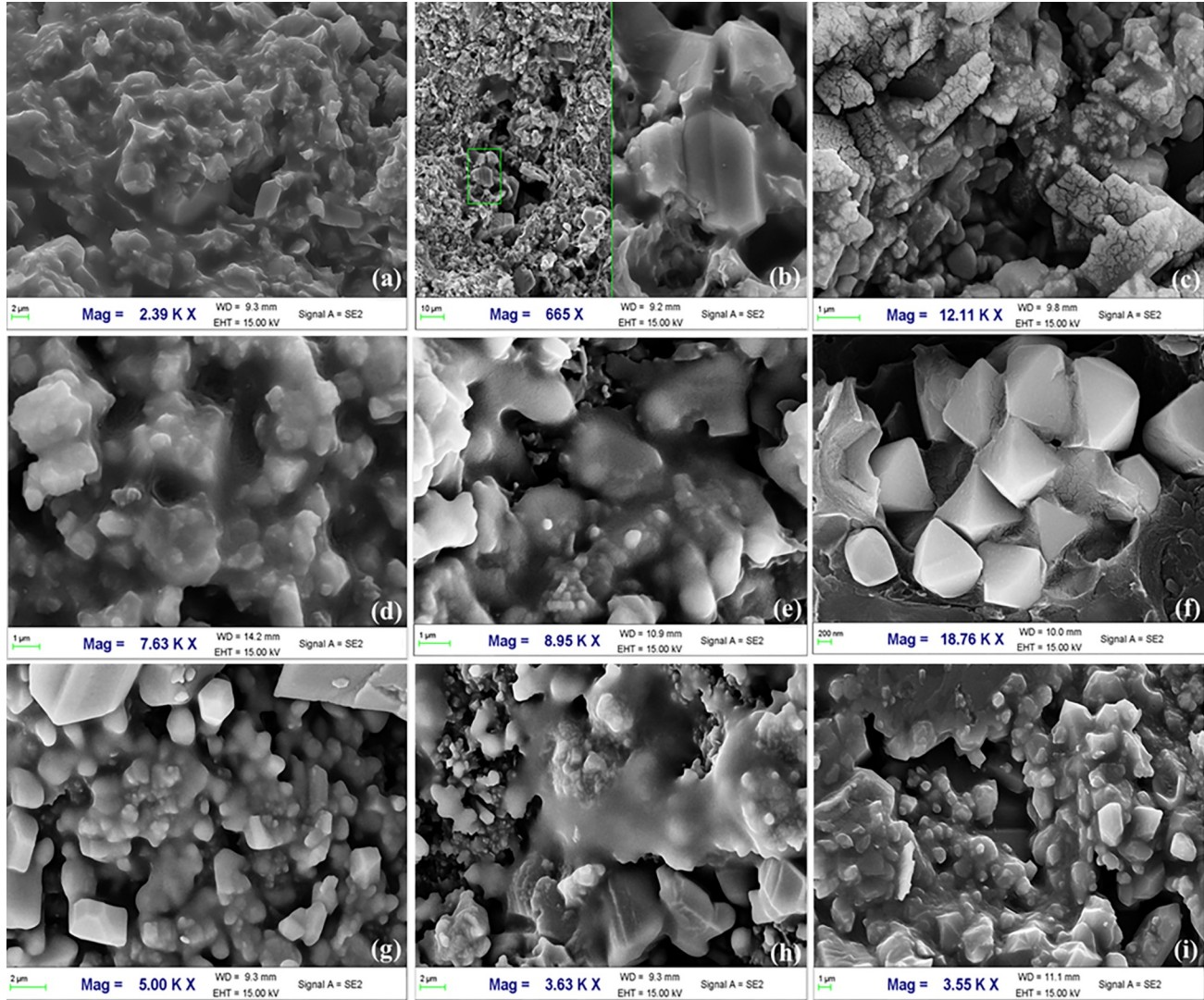

**Fig 5. Occurrence characteristics of soluble OM under SEM.** (a) $P_2l_1^{2-3}$, 3313.99 m, adsorbed oil on the surface of quartz and feldspar; (b) $P_2l_1^{2-3}$, 3313.99 m, dissolution pores and adsorbed OM on the surface of quartz and feldspar; (c) $P_2l_2$, 3526.82 m, minerals dissolution and the OM adsorbed on minerals; (d) $P_2l_2$, 3555.69 m, adsorbed oil on the surface of quartz and feldspar; (e) $P_2l_2$, 3559.77 m, intergranular pores and soluble OM; (f) $P_2l_2$, 3571.73 m, pyrite and quartz, and OM in intergranular pores; (g) $P_2l_2$, 3557.67 m, quartz, sodium feldspar and iron dolomite, and OM in micro-pores; (h) $P_2l_2$, 3533.18 m, quartz, sodium feldspar, dolomite dissolution, OM filled in the pores; (i) $P_2l_2$, 3549.29 m, quartz, sodium feldspar, OM filled in pores.

if it was a meaningful attempt, which can help us intuitively visualize the changes of different components during the extraction process. However, it should be noted that the residual samples tested were not identical, as the samples needed to be crushed to different degrees at each step in the extraction process. This means that the hydrogen-containing components in the samples, the size and the test sites of the samples varied during the experiment. Although 2D-NMR detection of samples before and after extraction did not provide continuous dynamic changes of hydrogen-containing components during extraction, it can still provide some reference for identifying different extracts.

The signals of the solid OM (region ①) were basically unchanged throughout the extraction process (Figs 3, 6 and 7), and they are the non-extractable component. The signal of hydroxyl/

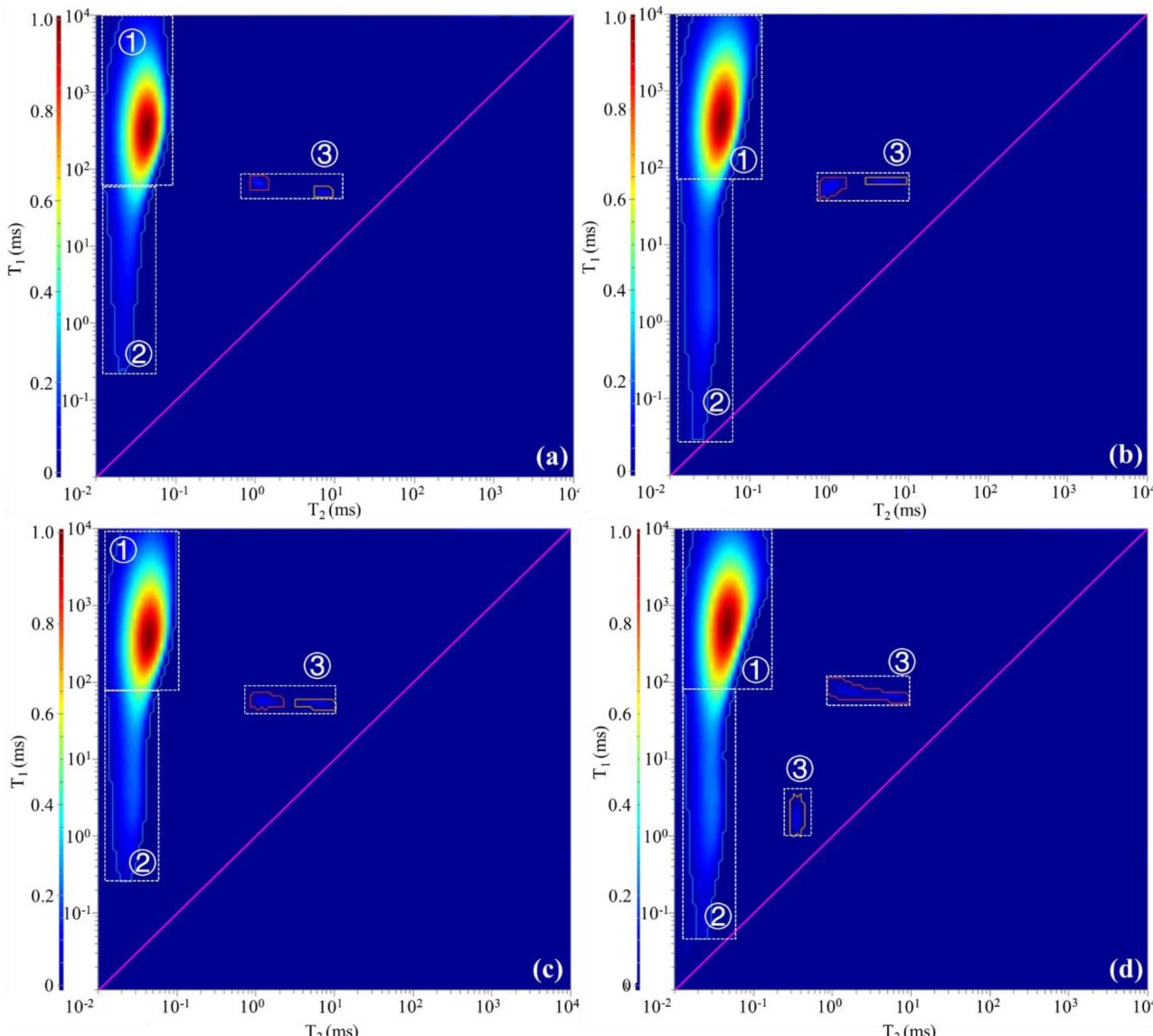

**Fig 6. Variation of hydrogen-containing components during the four-step extraction process for sample A.** (a)—(d) respectively represent the characteristics of hydrogen-containing components in the sample after the first extraction to the fourth extraction.

bound water (region ②) in sample B showed a large attenuation after the first extraction (Figs 3B and 7A), and no major changes happened in other samples (Figs 2, 6 and 7). Hydroxyl is a common polar group with the rational formula -OH. It has some similar properties to water and can form hydrogen bonds with water. Although some tightly structured kerogen molecules or heavy oil macromolecules also have -OH, they usually have larger $T_1$ and smaller $T_2$, which partially overlap with the solid OM signal in the $T_1$-$T_2$ diagram. In fact, the definition of region ② is influenced by the sample condition and human factors, and some oxygenated compounds rich in hydroxyl groups are challenging to distinguish. However, in actual exploitation, these components are difficult to extract. They are easily adsorbed on the surface of minerals, especially clay minerals, and even after repeated extraction with organic solvents, the adsorption layer cannot be completely extracted. Thus, the NMR signals of the hydroxyl-rich

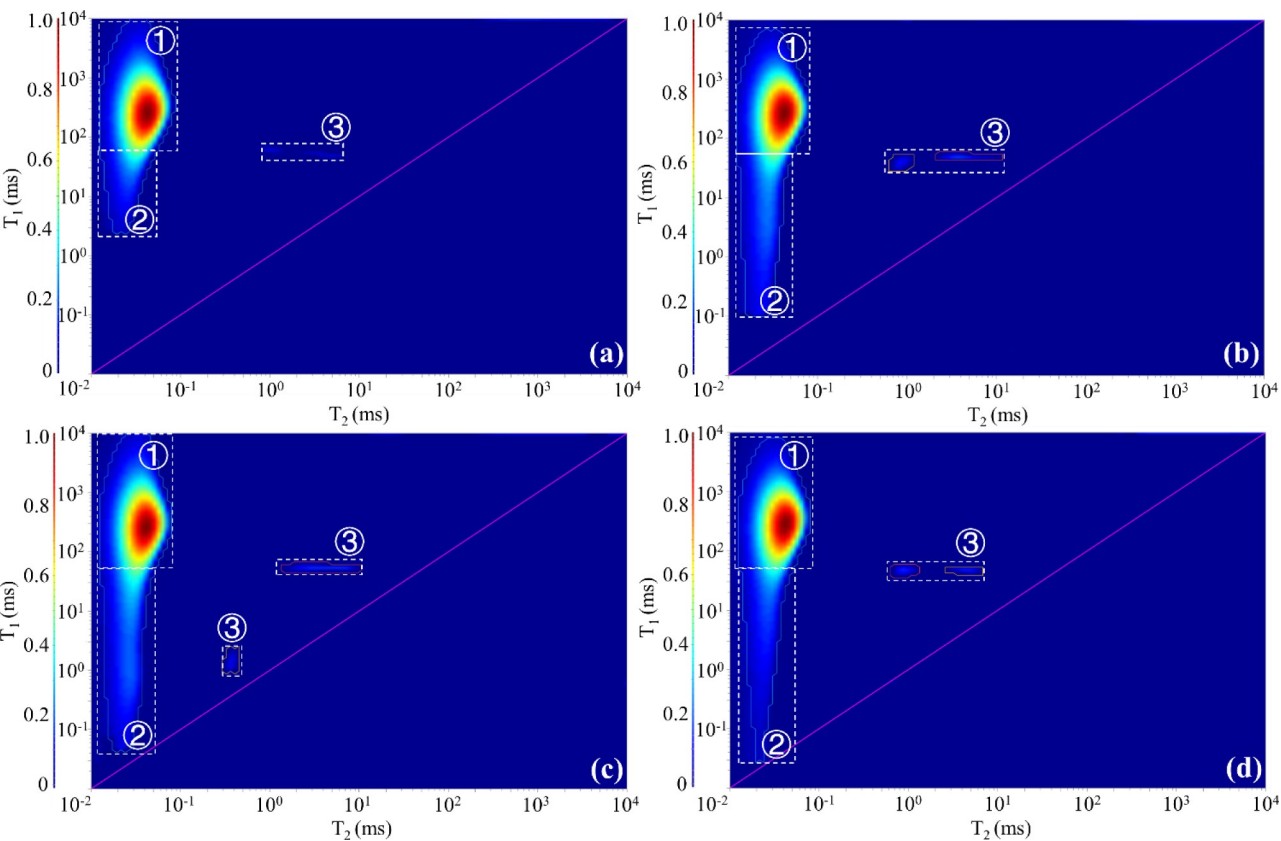

**Fig 7. Variation of hydrogen-containing components during the four-step extraction process for sample B.** (a)—(d) respectively represent the characteristics of hydrogen-containing components in the sample after the first extraction to the fourth extraction.

compounds (water or oxygenated organic compounds) represented by region ② are essentially unchanged throughout the extraction process. The NMR signal changes of soluble OM (region ③) are obvious (Figs 3, 6 and 7), but unfortunately, the extraction rate and residual amount of soluble OM could not be constrained quantitatively based on the NMR signal amount. Due to the variation in sample size and volume, the residual soluble OM content in the samples after each extraction step are not comparable and fluctuates irregularly.

After the four-stage extraction, soluble OM residues remained in the samples, which may be related to clay minerals and solid OM. We carried out the oil-washing experiments using the same samples and observed them under SEM. The results show that the residual OM is closely combined with the filamentous illite and chlorite, and attached to the mineral surface and the micro-pores formed by clay minerals (Fig 8). Clay minerals are important enrichment carriers for soluble OM. Unlike other clastic minerals, clay minerals possess a larger specific surface area [58]. Most clay minerals have a layered structure [59] and OM can be preserved by entering the interlayer domain of clay minerals to form a stable organic-clay complex [60].

## 4.3 The occurrence types of different extracts

In multi-stage extraction, particle size is an important factor affecting the extraction efficiency [17, 61]. The crushing process can achieve the following objectives: ① increase the specific surface area of the samples, ② release isolated pores, ③ break the interfacial tension between the solid and the liquid, ④ promote closer contact between the solvent and the samples [61, 62].

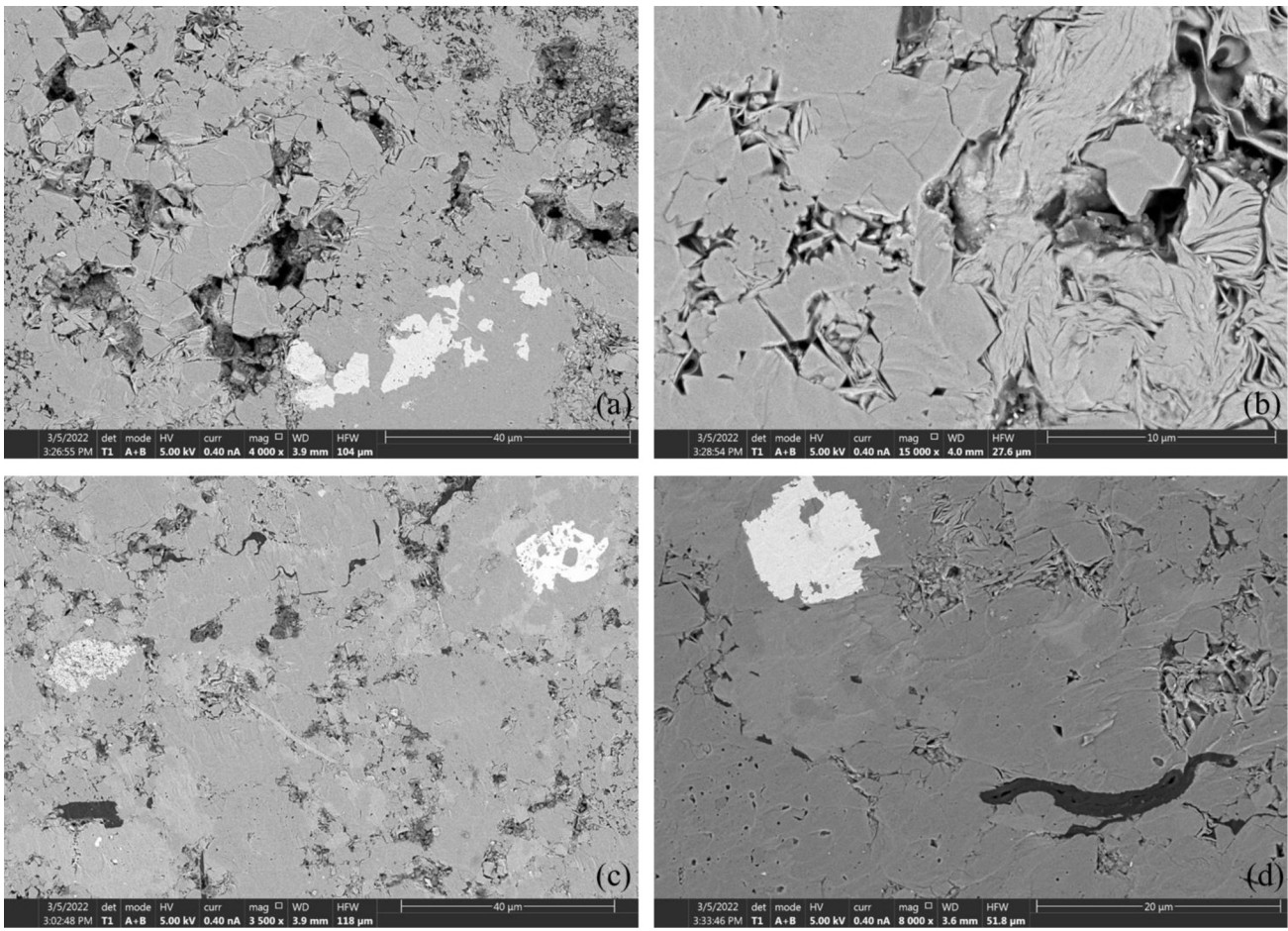

**Fig 8. Residual soluble OM in samples A and B after oil washing, residual OM is closely combined with the filamentous illite and chlorite.** (a) and (b) represent Sample A, while serial numbers (c) and (d) represent Sample B.

Assisted by different polar solvents, soluble OM in different occurrence states can be progressively extracted, e.g. free oil, adsorbed oil, and inclusions oil [26] (Table 1). The particle size and solvent selected in our experiment can reflect the main migration mechanisms of different occurrence states of shale oil, as well as the adsorption and desorption processes. Note that multi-stage extraction does not achieve complete separation of shale oil from different occurrence states [26], but it does reflect the geological regularity as much as possible [17, 63].

Extract I was obtained by extracting block samples (about 1cm in size) using weakly polar organic solvents. The content of saturated hydrocarbons in extract I is the highest, followed by non-hydrocarbons, while aromatics and asphaltenes are the least (Fig 9), which can be considered as free oil. The mobility of the oil, obtained during the depletion production stage, was the strongest as the energy (temperature, pressure) of the stratum itself is enough. This part of the crude oil can be stored in the pore system between the mineral skeleton and enter the wellbore along the artificial fractures formed by hydraulic fracturing to form the initial product, which is considered free oil in production. We obtained the group component of crude oil data during the self-spray stage from our partner. The results show that the components of shale oil in the self-spray stage were mainly saturated hydrocarbons, with a percentage content above 40%: minimum, 41.40%, maximum, 66.76%, and average, 56.33%. The content of aromatic hydrocarbons was low, with a minimum value of 10.51%, a maximum of 20.91%, and a

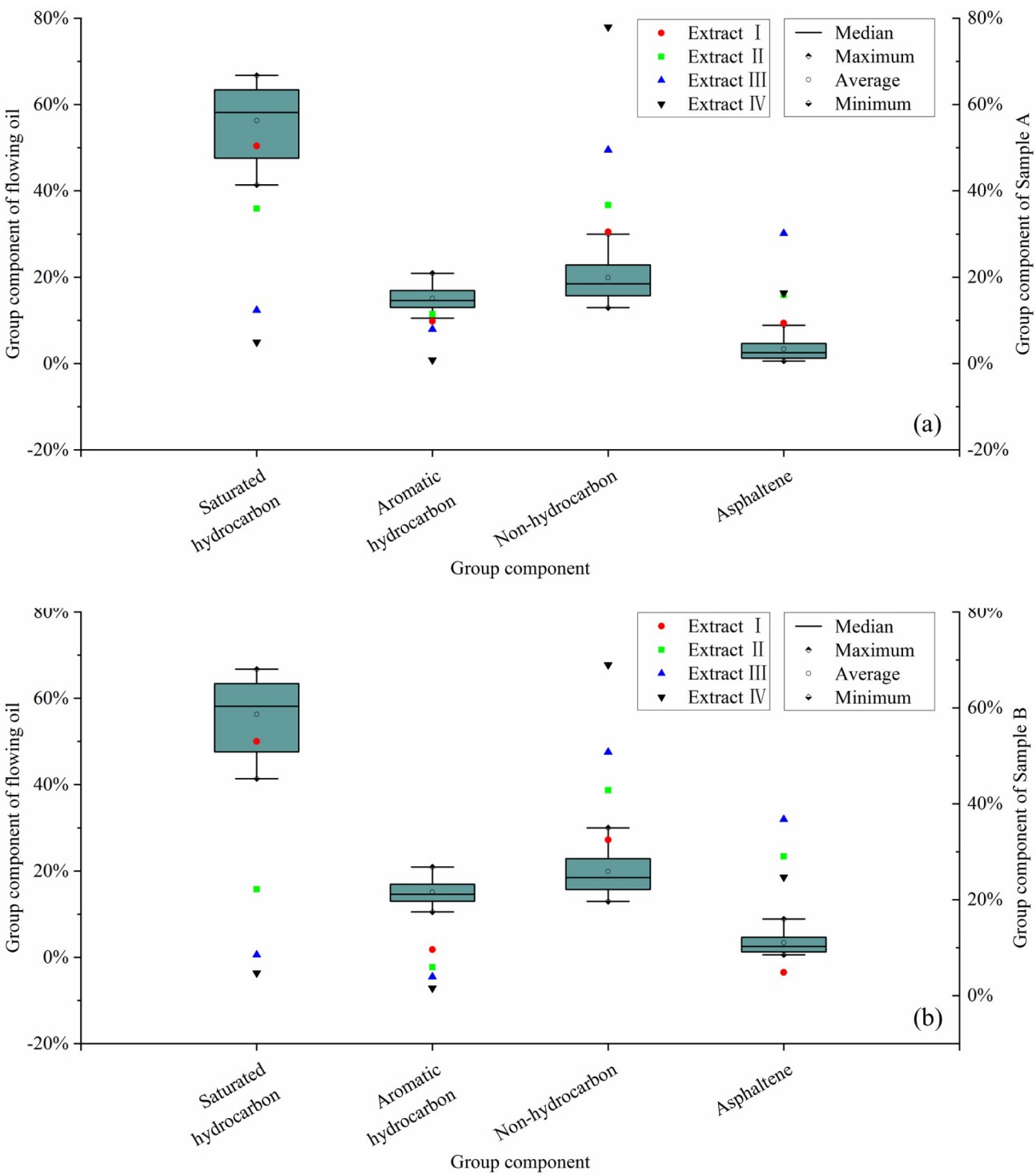

**Fig 9. Comparison of hydrocarbon group composition characteristics between extracted compounds from samples A and B after four-step extraction and crude oil at the self-spray stage.** (a) The comparative result for Sample A; (b) The comparative result for Sample B.

mean of 15.10%. The average value of non-hydrocarbon was 19.92%, with a minimum value of 12.93 and a maximum of 30.00%. The asphaltene content was the lowest, with a mean value of 3.34%, a minimum of 0.58%, and a maximum of 8.85% (Fig 9). We further compared the group components of the extracts of samples A and Sample B at different stages with the crude oil at the self-spray stage. The results show that the extract I is closest to the crude oil in the self-spray stage (Fig 9). Therefore, it is consistent with the production law to define extract I as free oil.

In the second extraction step, the samples were further crushed to about 0.1 cm, which simulated the large-scale fracturing process during exploitation. Fracturing can create artificial fractures. Fracturing fluids containing surfactants can alter the surface relationship between crude oil and rocks and activate some shale oil with weak fluidity. Therefore, extract II also belongs to the free oil, whose group components are also similar to the crude oil in the self-spray stage. However, extract II was not as mobile as the free oil in the first extraction step. The content of non-hydrocarbon and asphaltene is markedly increased, and the content of saturated hydrocarbons is lower than that in extract I (Fig 9). The differences in group components can affect the viscosity of crude oil, thereby affecting its mobility. The viscosity of crude oil is inversely proportional to its saturated hydrocarbon content and positively proportional to its non-hydrocarbon and asphaltene content. Therefore, the mobility of Extract II is not as good as Extract I.

When the samples were crushed to 150 mesh (about 95 μm), the micron-sized storage spaces of the samples were largely destroyed, and most mineral particles were fully separated. Therefore, the solvents could better contact and dissolve the residual OM. The third step used the same combination of weakly polar organic solvents as in the previous two steps, which allows further stripping of the small and medium molecular weight soluble OM within the network structures on the mineral surface and the inner and outer surfaces of kerogen frameworks. While the fourth step used a combination of organic solvents with stronger polarity, the OM with polar groups adsorbed on the surface of minerals and kerogen can be extracted. Therefore, extract III and extract IV may be considered adsorbed shale oil. However, from a production perspective, there is a substantial difference in their group composition compared to the crude oil produced during the self-spray stage. The most remarkable difference is that the content of saturated hydrocarbons is extremely low, and the content of non-hydrocarbons and asphaltenes is significantly increased compared to free oil (Fig 9). Moreover, compared with extract III, extract IV has a higher non-hydrocarbon content, reaching 70%-80%; the asphalt content is around 20%; the saturated hydrocarbons and aromatics content is low, less than 10% and 2% respectively (Fig 9). Predecessors consider that these differences are related to molecular adsorption effects and maturity of the oil. Early-charged crude oil often has higher heavy components and lower aliphatic content, and its maturity is lower than that of late-charged crude oil [26, 64].

We did not obtain the crude oil data for the entire production cycle. Hence, we cannot give an accurate definition of extracts III and IV from a production perspective. The adsorbed state may be the major form of their occurrence, but other forms of occurrence cannot be ruled out, such as the inclusion oil, dissolved oil, and the oil in the interlayer domain. Therefore, we suggest it is more accurate to call this part of soluble OM residual oil, which is the crude oil that cannot be extracted with the current technology but can be recovered in the future once a breakthrough in technology emerges.

## 4.4 Micro-occurrence mechanism

**4.4.1 Differentiation effects.** Laser confocal SEM (LSCM) integrates microscopy, high-speed laser scanning, and image processing technology. Combined with a fluorescence

microscope, the occurrence characteristics of different components of shale oil in the original rock samples can be well depicted, and a more intuitive 3D occurrence model of shale oil can also be established.

The results of LSCM indicate that the hydrocarbons in the samples have undergone significant differentiation effects, as the light and heavy components are unevenly distributed (Fig 10). In the 3D model, the heavy component in red is located on the outside and the light component in blue is inside and wrapped by the heavy component. The light and heavy

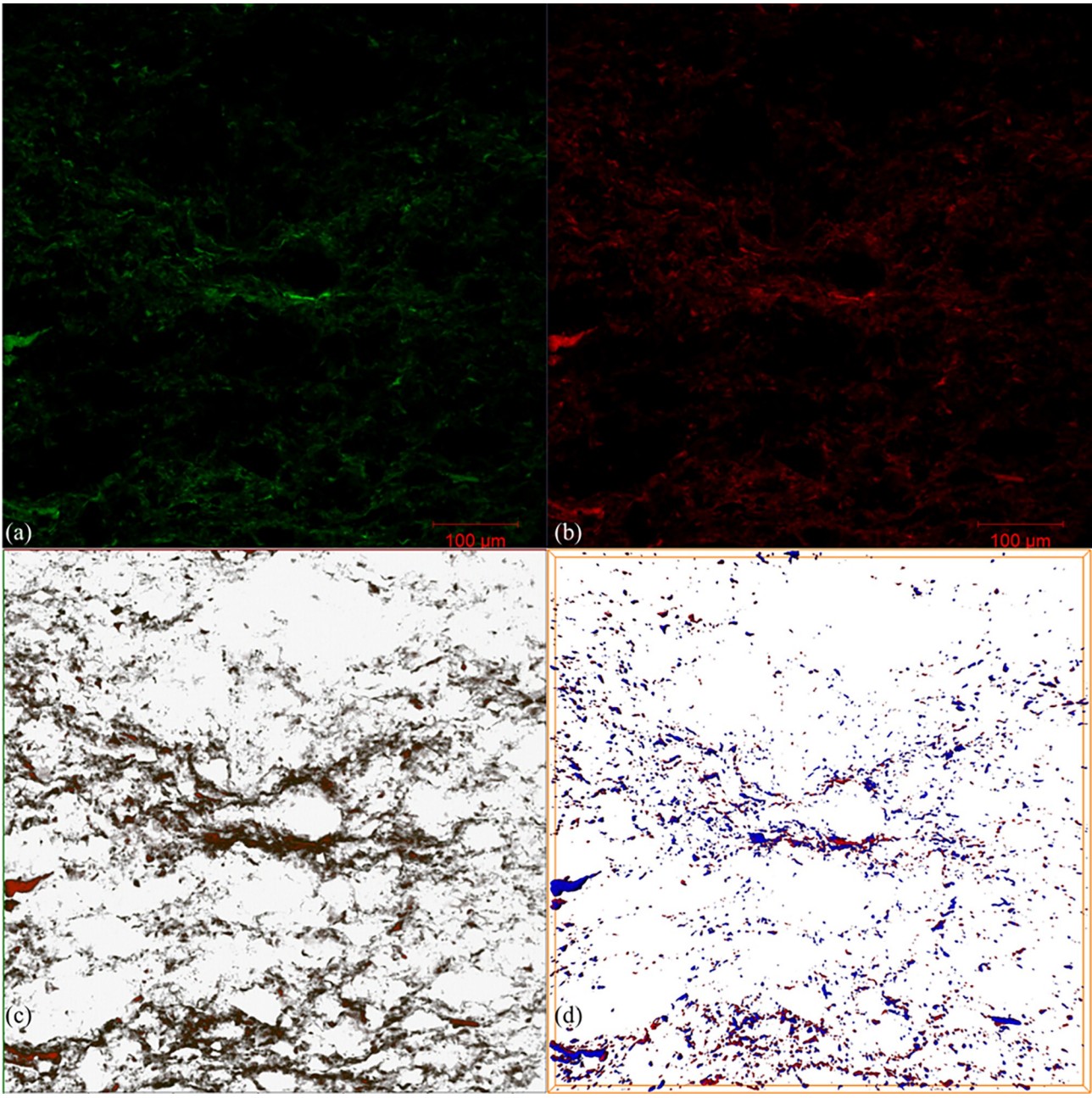

**Fig 10. Laser confocal Microscopy (LSCM) results.** (a) Light component laser confocal 2D scanning result; (b) Heavy component laser confocal 2D scanning result; (c) Synthesized 2D image of light and heavy components. Green is the light component, red is the heavy component; (d) Synthesized 3D image of light and heavy components. Green is the light component, red is the heavy component.

**Table 2. Organic functional group band of the infrared spectrum.**

| Wavelength (μm) | Wave number(cm$^{-1}$) | Chemical bond type |
|---|---|---|
| 2.7~3.3 | 3750~3000 | $\upsilon_{OH}$, $\upsilon_{NH}$ |
| 3.0~3.3 | 3300~3000 | $\upsilon_{CH}$ (-C≡C-H, C = C, Ar-H) |
| 3.3~3.7 | 3000~2700 | $\upsilon_{CH}$ (-CH$_3$, -CH$_2$-, C-H, H-C = O) |
| 4.2~4.9 | 2400~2100 | $\upsilon_{C≡C}$, $\upsilon_{C≡N}$, $\upsilon_{-C≡C-C≡C-}$ |
| 5.3~6.1 | 1900~1650 | $\upsilon_{C = O}$ (acid, aldehyde, ketone, amide, ester, estolide) |
| 6.0~6.7 | 1680~1500 | $\upsilon_{C = C}$ (aliphatic and aromatic), $\upsilon_{C = N}$ |
| 6.8~7.7 | 1475~1300 | $\delta_{C-H}$ (in-plane stability), $\upsilon_{X-Y}$ |
| 10.0~15.4 | 1000~650 | $\delta_{C = C-H}$, $\delta_{Ar-H}$ (in-plane stability) |

components are well delineated, showing a clear differentiation phenomenon (Fig 10D). The light component content is 1.59% (v/v) and the heavy component content is 1.41% (v/v); the light-to-heavy ratio is 1.13. The differentiation of light and heavy components in micro-pores is formed with long-term geological history, similar to the geological color layer effect in the process of oil migration. However, in the shale oil system of the Lucaogou Formation, crude oil only migrated over a short distance, or not at all [65, 66]. After entering the pores, crude oil interacted with minerals. Due to the influence of crude oil components and mineral surface properties, the wettability of mineral surfaces gradually changes, resulting in the differentiation of light and heavy components.

**4.4.2 Oil-rock interaction.** The essence of the differentiation between the light and heavy components is the interaction between oil and rock. The major influence factors are the properties of the organic functional groups and the surface properties of minerals. OM and minerals have strong chemical activity, which can be combined through hydrogen bonds, ion-dipole, electrostatic interaction, and van der Waals force [67].

Infrared spectroscopy yields an absorption spectrum for us to study the interaction between infrared light and substance molecules. It is widely used in the structural analysis of organic compounds and is an important tool for determining the functional groups of the compounds. The wave number of infrared spectra in this test is 4000~400 cm$^{-1}$ and the wavelength is 2.5~25 μm, which can reveal the fundamental vibration frequencies of most organic compounds. Table 2 lists several regions that are key to the identification of the functional groups of organic compound molecules.

The infrared spectra of extracts of samples A and B with different occurrence states are shown in Fig 11. The components of extract I and extract II are mainly aliphatic hydrocarbons, which are distributed continuously from C$_{13}$ to C$_{30}$, with main peaks (Fig 12). The functional groups are mainly -CH$_2$- and -CH$_3$ (Fig 11A, 11B, 11E and 11F). Extract III contains hydrocarbons and heteroatomic compounds. The functional groups of hydrocarbons include aliphatic hydrocarbons (-CH$_2$-, -CH$_3$) and aromatic hydrocarbons (-C$_6$H$_5$), as well as others. The functional groups of the heteroatomic compounds are nitrogen (-N-), hydroxyl (-OH), phenol group (-C$_6$H$_5$O), ether group (-O-), and ester group (-COO-), and one or more functional groups are contained in the same molecule (Fig 11C and 11G). We detected a variety of functional groups in extract IV, including hydroxyl (-OH), ether group (-O-), carboxyl (-COOH), and ester (-COO-), as well as ethylene linkage (= C = C =) and aromatic ring (-C$_6$H$_5$) (Fig 11D and 11H). The vast majority of the molecules contain more than two functional groups.

Previous studies on mineral-water interfaces suggested primarily four different types of functional groups on mineral surfaces: hydroxyl-type functional groups, Lewis acid sites (or Bronsted acid sites), salt (or sulfur) groups, and surface constant charge. Among them, the hydroxyl-type surface functional groups are the most basic group, which exists on the surface

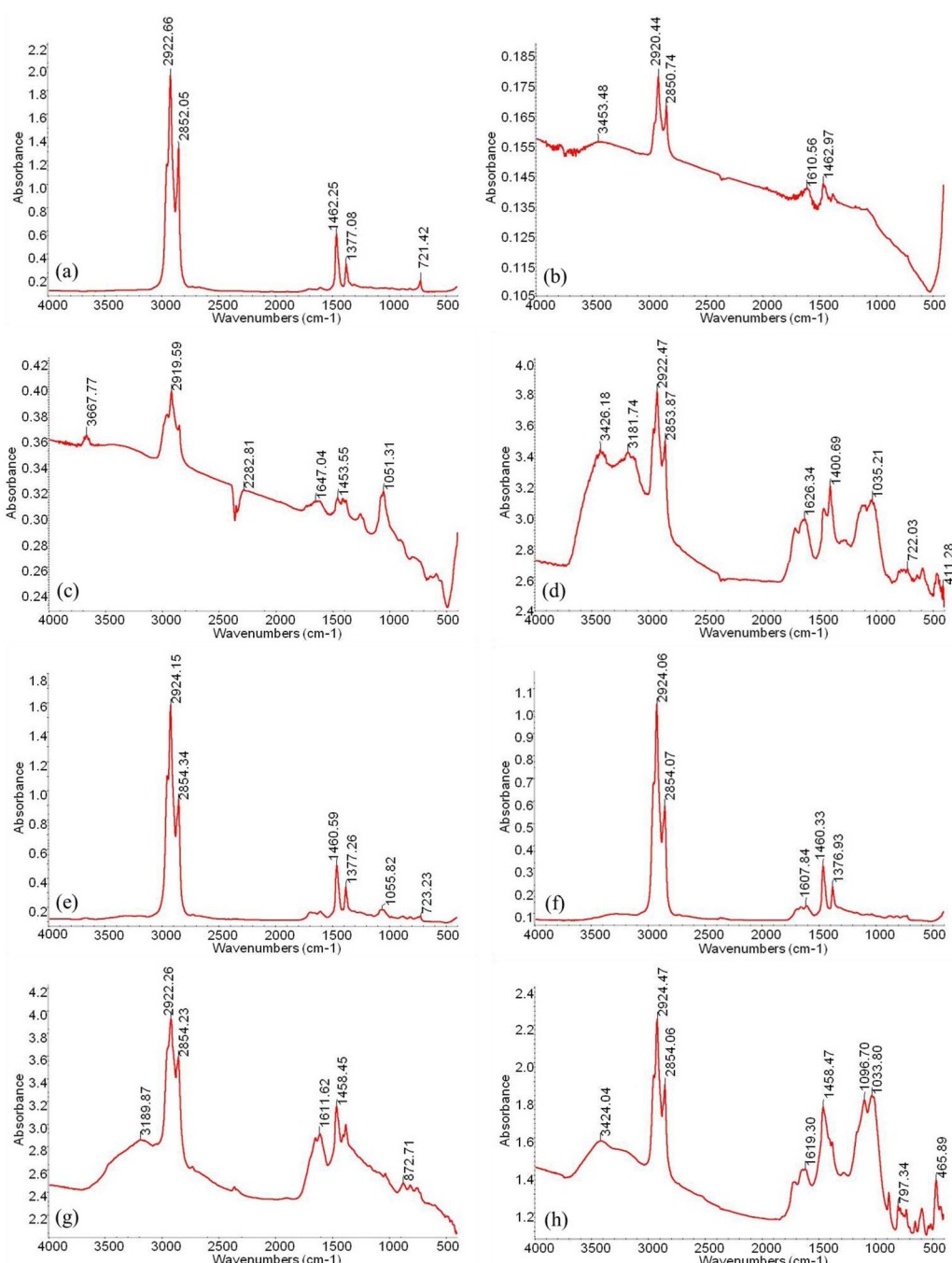

**Fig 11. Infrared spectroscopy results of extracted compounds from four-step extraction of sample A and sample B.**
(a)—(d) Four-step extraction products of sample A; (e)—(h) Four-step extraction products of sample B.

of various minerals [68–70]. The adsorption of natural OM also occurs mainly at hydroxyl sites on mineral surfaces. Components with high molecular weight are easily adsorbed because they have more aromatic carbon content and acidic functional groups than the smaller molecular components [71–73]. The functional groups of heteroatomic compounds in extracts III and IV, i.e. nitrogen (-N-), hydroxyl (-OH), phenolic (-C_6H_5O), ether (-O-), and ester (-COO-) groups, can easily form adsorption layers through surface coordination or complexation with

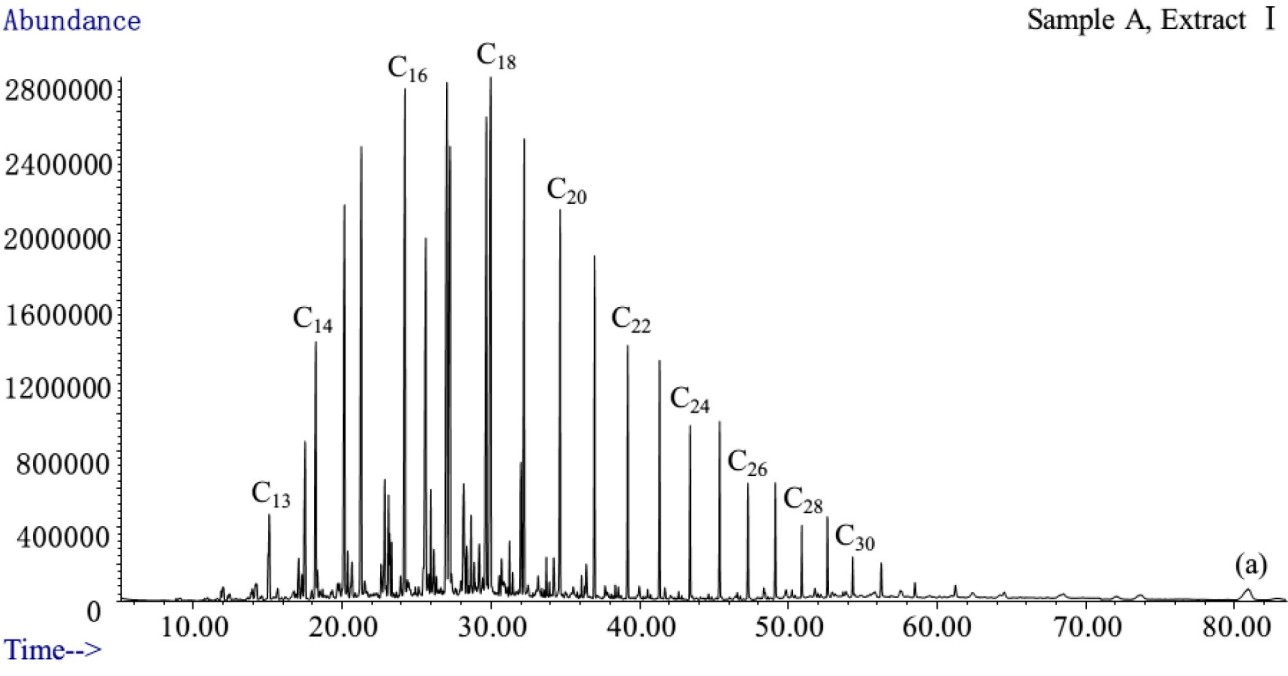

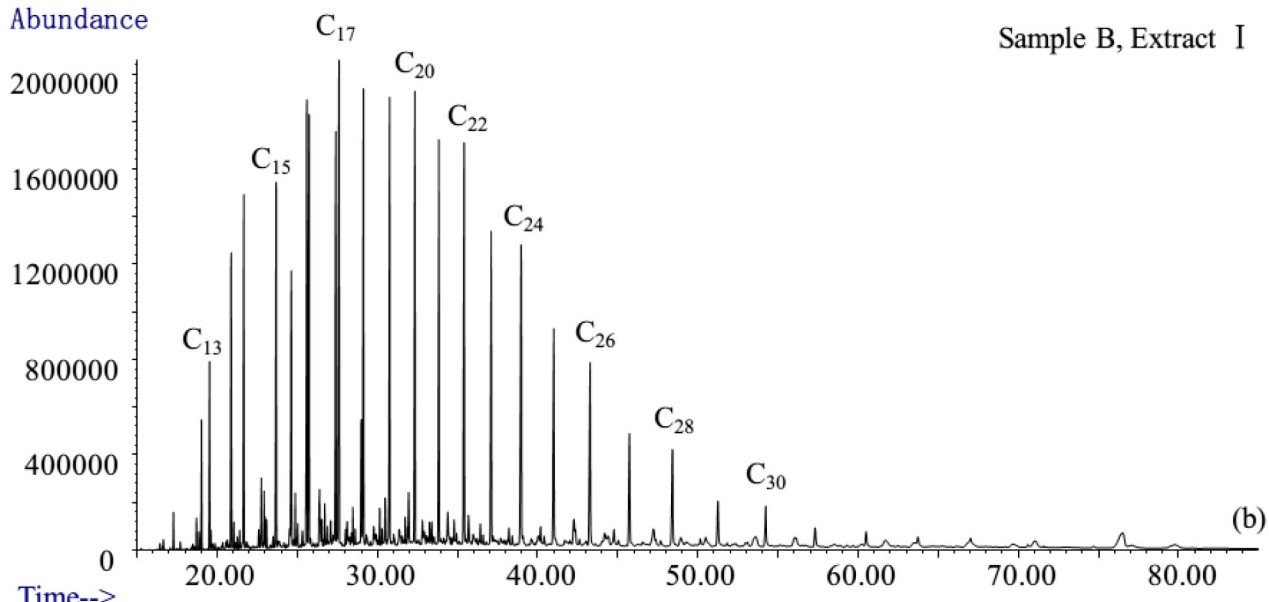

**Fig 12. Chromatograms of the first step extracts of sample A and sample B.** (a) Result of Sample A; (b) Result of Sample B.

the hydroxyl groups on the mineral surface. Clay minerals have stronger adsorption properties than quartz, feldspar, and other minerals, and are very easy to adsorb organic molecules. Soluble OM either exists in the ultramicroscopic pores of the clay surface by physical adsorption or penetrates deeply into the interlayer domains within the clay minerals through chemisorption, forming an organic-clay complex [60]. In addition to these OSN compounds, extracts IV and III are also rich in asphaltenes which belongs to polar substance. Asphaltenes are easily adsorbed on the mineral surface leads to mineral show lipophilicity, and their adsorption is

strong, making it difficult to remove using conventional oil-washing methods. This means that it is extremely challenging to use the residual oil in the actual exploitation process [74].

The main components of free oil are aliphatic hydrocarbons. Aliphatic hydrocarbons are the main components of crude oil and are mainly composed of $-CH_2-$ and $-CH_3$ functional groups. The adsorption of free oil on the surface of rocks and minerals is much weaker than that of the residual oil, but some long-chain alkanes can also adhere to the surface of minerals or asphaltenes, forming a weak adsorption layer. The length of the carbon chain is the main factor affecting the adsorption of free oil. For single-component alkanes, high carbon number alkanes have more adsorption tendency and tone to aggregation on solid surfaces. The more the number of $-CH_2$, the stronger the adsorption tendency [75, 76]. Therefore, some free oils with long-chain alkanes and residual oil that are rich in polar substances and OSN compounds (i.e., extracts III and IV) will form multiple adsorption layers in the region near the pore wall. The closer to the minerals, the higher the density of the adsorption layer and the more tightly it is adsorbed. Also, compared to large pores, small pores are more likely to cause the adsorption of polar components due to the filling effect [77].

The tests of the organic element in samples A and B show that the content of N and O elements in the residual oil is higher than in free oil. From extracts I to IV, the content of N and O gradually increases, and C, H, and H/C ratio gradually decreases while O/C ratio increases (Table 3). Previous studies have unraveled that mineral surfaces are preferentially enriched with OSN [78]. The residual oil in the pore system may be stratified according to the type and polarity of the OSN, showing moderate and strong adsorption layers, respectively [79]. Moreover, heterocyclic compounds with N in crude oil will replace cations through the cation exchange effect, directly adsorbed on the negatively charged silanol group ($\equiv$SiO-) on the surface of the sandstone. Therefore, the increase of N and O will promote the reversal of wettability from water to oil, which is beneficial for the adsorption of the polar heavy components.

Overall, the oil-rock interaction determines the micro-occurrence characteristics of shale oil, i.e. location, content, and composition. Residual oil is mainly adsorbed on the inner and outer surfaces of minerals, which is related to the polar heavy components, i.e. OSN compound groups, polar substances, and large molecular weight in the residual oil. Asphaltenes and non-hydrocarbons are more easily adsorbed on mineral surfaces by ionic or hydrogen bonds than aliphatic hydrocarbons [74]. Therefore, the components of residual oil show a high content of asphaltene and non-hydrocarbon components and a low content of saturated hydrocarbons and aromatics. While the free oil is dominated by non-polar hydrocarbons. The free oil mostly enriched inside the pore space and partially attached to the surface of quartz, feldspar, or heavy components of early charge. Therefore, as the extraction proceeds, the order in which free and adsorbed shale oil is extracted should be a reverse-charged process. From

**Table 3. Elemental content of carbon, hydrogen, oxygen, and nitrogen in different extracts.**

| Samples | No. | Element content (%) | | | | H/C | O/C |
|---|---|---|---|---|---|---|---|
| | | N | C | O | H | | |
| A | Extract I | 1.25 | 86.08 | 2.49 | 12.63 | 1.76 | 0.02 |
| | Extract II | 3.74 | 75.56 | 6.41 | 9.66 | 1.53 | 0.06 |
| | Extract III | 4.02 | 50.86 | 6.43 | 6.04 | 1.42 | 0.09 |
| | Extract IV | 5.63 | 57.00 | 33.95 | 7.53 | 1.59 | 0.45 |
| B | Extract I | 2.39 | 85.49 | 2.36 | 11.87 | 1.67 | 0.02 |
| | Extract II | 2.18 | 85.41 | 2.73 | 10.81 | 1.52 | 0.02 |
| | Extract III | 3.83 | 83.86 | 3.75 | 9.19 | 1.32 | 0.03 |
| | Extract IV | 5.69 | 51.96 | 20.42 | 6.66 | 1.54 | 0.29 |

the free oil to the adsorbed oil, the distance from the mineral surface decreases and the oil-rock interaction strengthens. The proportion of the saturated and aromatic hydrocarbons gradually decreases, while the proportion of the non-hydrocarbons and asphaltenes increases (Fig 9). These differences are also closely related to the maturity of OM and the process of hydrocarbon accumulation. And ultimately be reflected through the micro-occurrence state of shale oil.

### 4.5 Shale oil accumulation and micro-occurrence model

The main accumulation pattern of the Lucaogou Formation is that oil only originated from interbedded mudstones without mixing. Crude oil generally comes from source rocks interbedded with reservoirs, without an obvious migration process [80]. It is generally suggested that the original reservoir was hydrophilic before being charged with oil and gas, and the mineral surface was wetted by water. Formation water is widely distributed within the pores and adsorbed on the minerals. With the maturity of OM, some organic substances in oil were adsorbed onto the mineral surface when the oil was charged into the reservoir, resulting in a reversal of the reservoir wettability [81].

Based on the present-day maturity of the source rocks in the Lucaogou Formation, with $R_o$ = 0.9% [82], the burial history can be classified into three stages: immature ($R_o$<0.5%), early maturity ($R_o$ = 0.5%~0.7%), and middle maturity ($R_o$ = 0.7%~1.1%). Formation water participated in the thermal evolution of kerogen and began the immature stage. However, the energy generated by thermal degradation was insufficient to break down the complex macro-molecular organic matter, thus primarily producing a small amount of soluble asphalt during this stage. In the early maturity stage, formation water was further depleted. Kerogen pyrolysis generated heavy oil with low maturity, which gradually charged the reservoirs and displaces the residual pore water under the pressure generated by hydrocarbon generation. OSN compounds and polar substances (asphaltenes) in heavy oil underwent coordination or complexation with mineral surfaces, were gradually adsorbed onto minerals, and ultimately, caused a reversal of the reservoir wettability from water wetting to neutral and oil wetting [83]. The medium maturity stage is the current maturity of the hydrocarbon source rocks of the Lucaogou Formation. Non-polar hydrocarbons are produced in large quantities in this stage. These hydrocarbons are the main components of free oil, and their adsorption is mainly related to the length of the carbon chain due to the dehybridization atom reaction. They exist mainly inside the pores, and partly adhere to the low-maturity component's surface, forming a weak adsorption layer (Fig 13).

## 5. Conclusions

Free oil and residual oil are the main occurrence states in the Lucaogou Formation in the Xinjiang oilfield, NW China. Free oil is mainly stored in pores and cracks and is the main target in current exploitation. Residual oil tends to adhere to the mineral surfaces and has selective adsorption, which is preferentially adsorbed on solid OM and clay particles. Free shale oil is a product of late charged, and aliphatic hydrocarbons are its main components. Residual oil is rich in non-hydrocarbon components and polar heavy components and is a product of early charged.

Oil-rock interaction is the driving factor that controls the micro-occurrence characteristics of shale oil. The interaction mode and binding strength between crude oil and minerals are influenced by their respective chemical activities. Polar substances in residual oil, e.g. asphaltene, and OSN compounds, are easily adsorbed on minerals surface. The accumulation of polar substances can invert the reservoir wettability from water to oil wettability. Different

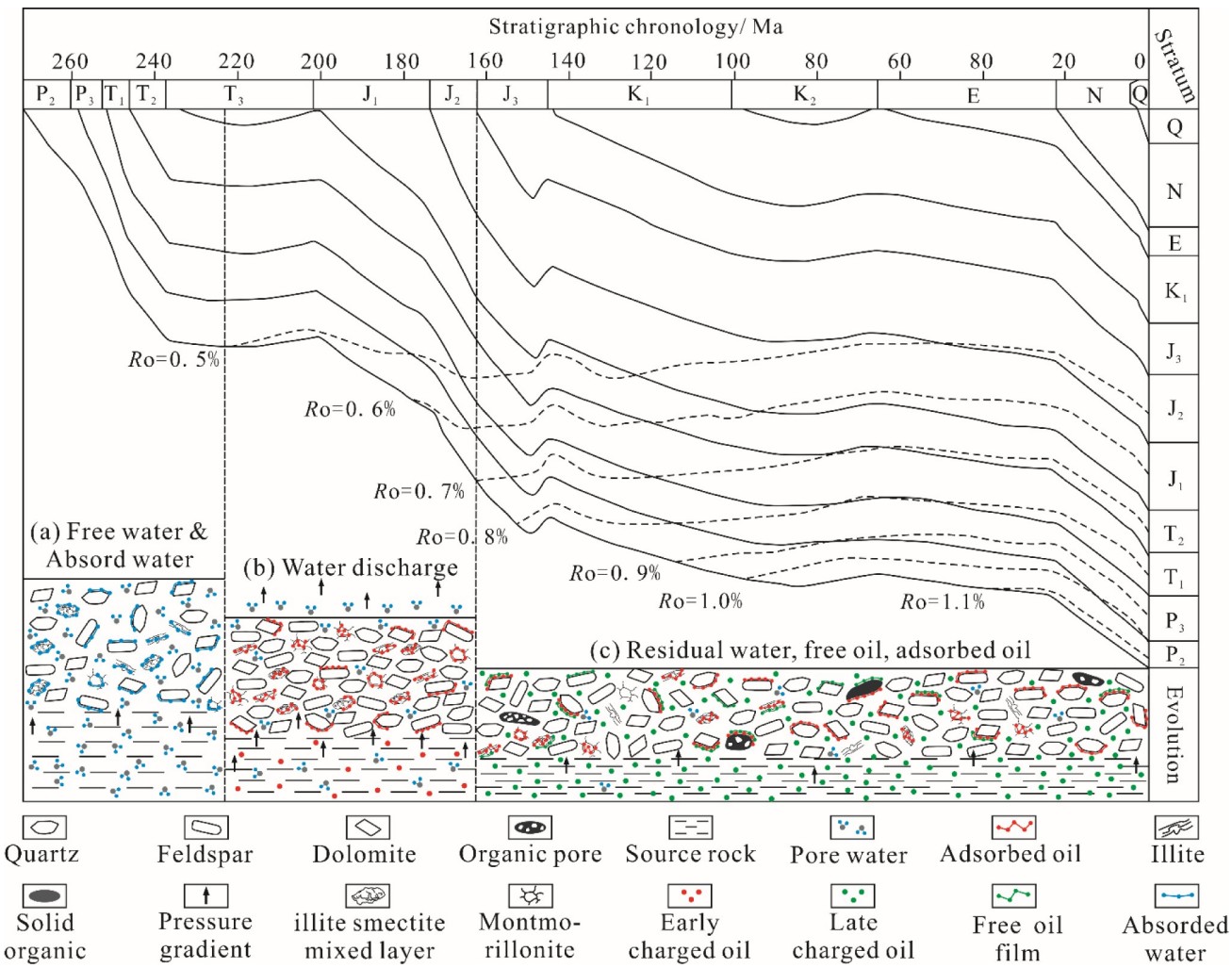

**Fig 13. Filling history [84] and micro-occurrence model of shale oil in the Lucaogou Formation.**

minerals have different types of surface interactions and different adsorption abilities. The special layered structure and huge surface area of clay minerals can attract crude oil to form organic-clay complexes through surface coordination or complexation. The adsorption capacity of the free oil is related to the aliphatic hydrocarbon carbon chain length. Part of the free oil can adhere to the surface of the adsorbed oil to form a weak adsorption layer, while the later charged free oil is present inside the pores, with the strongest mobility.

The Lucaogou Formation is rich in complex hydrogen-containing components. Although soluble OM can be extracted through organic solvents under laboratory conditions, they cannot be fully extracted in actual exploitation. Free oil separated by multi-stage extraction is currently the main object of exploitation and is mostly utilized in the self-spray stage. The utilization of residual oil is poor. Although multi-stage large-volume fracturing technology can create a sizable fracture network, and fracturing liquid containing surfactants can activate some residual oil, it is not yet known whether the residual oil corresponds to the third or fourth stages of the extraction experiment can be exploited. In future studies, crude oil samples from the entire production cycle should be collected and compared with laboratory extracts to clarify the meaning of residual oil with production guidance.

## 6. Prospects

The Lucaogou reservoir exhibits extremely low matrix permeability. Consequently, the effective exploitation of this reservoir necessitates the implementation of production methods such as horizontal drilling and volume fracturing to create artificial fracture systems. The shale oil system is currently in its initial stage of exploitation, benefiting from the substantial energy reservoir within the stratum. Presently, the primary production capacity is achieved through well self-spray operations, with pressure control managed via nozzles. Typically, the self-spray phase of most wells spans approximately 3 years. At the outset of self-spray production, the fracturing process plays a critical role in determining the production capacity of horizontal wells [85]. Currently, the horizontal wells fracturing and high-strength volume fracturing technology used in the work area have reached better effect [86]. However, there is a general problem of production decline in the Lucaogou Formation, with a decline rate of 33–65% during the self-spray stage and 16% after the pump swabbing stage. One million tons can only maintain a stable production for 5 years [87].

According to this study, the shale oil exploited in the self-spray stage is mainly free oil. Although some horizontal wells have increased production by increasing soak time after fracturing, a large amount of residual oil is still difficult to be exploited. Production by pumping after the self-spray stage not only increases the costs, but also makes it challenging to stabilize the yield due to the impact of the amount of liquid supplied. Adequate liquid supply can maintain low oil yield, but once insufficient, the yield decreases sharply. Theoretically, part of the residual oil can be exploited in the pump swabbing stage, but due to the lack of crude oil data throughout a full production cycle, it is uncertain whether the residual oil in the three or four stages of the extraction experiment can be exploited (Fig 14). Therefore, in terms of theoretical research, it is essential to enhance the investigation of shale oil adsorption and desorption mechanisms under real stratum conditions. Utilizing cryo-electron microscopy at ultra-low

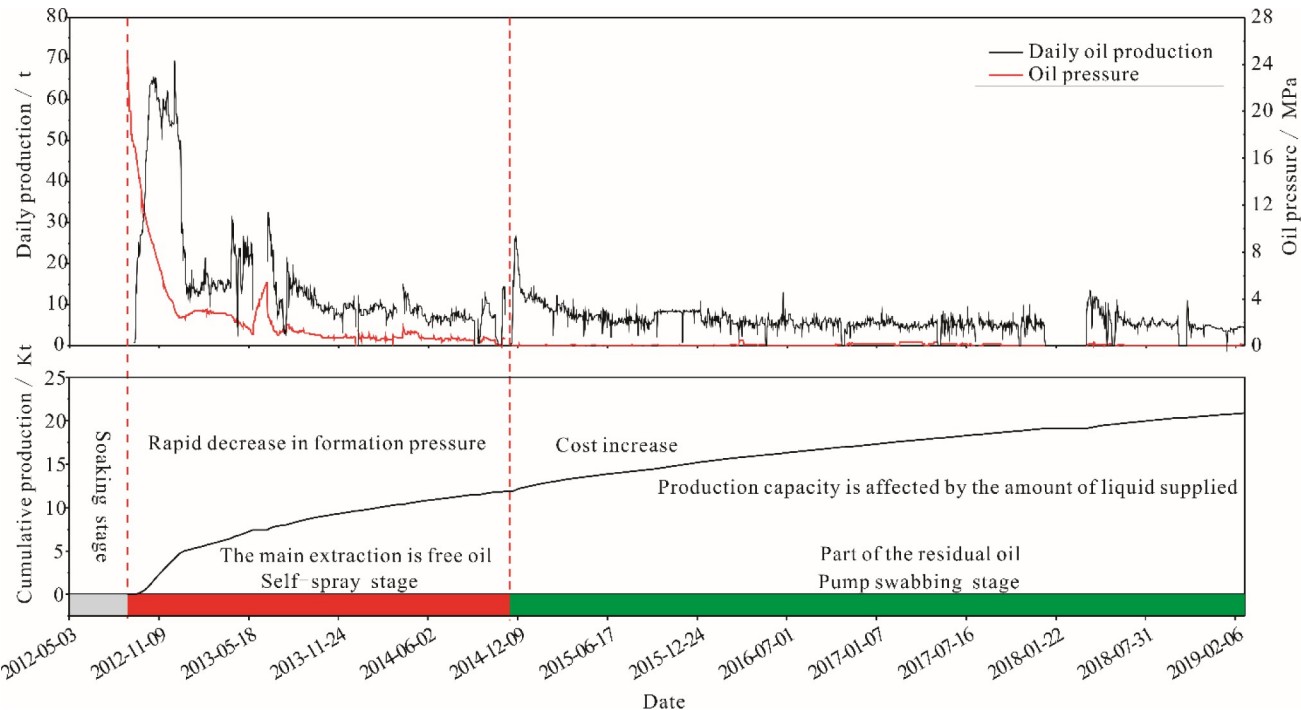

**Fig 14. Production dynamics of a horizontal well from the soaking stage to the pump swabbing stage.**

temperatures, in-depth observations of fresh core samples should be conducted to provide a visual representation of the micro-occurrences of shale oil.

In the context of exploitation, carbon dioxide fracturing and displacement technology is a potential solution for increasing production. $CO_2$ in the supercritical state has the dual characteristics of gas and liquid, with strong permeability and solubility. It has good solubility and strong extraction ability in crude oil, which can diffuse, dissolve, extract, and mix with crude oil, reducing oil viscosity and interfacial tension [88, 89]. During the injection stage, $CO_2$ can enter the matrix to increase stratum energy, mix with crude oil to reduce viscosity, and improve the crude oil fluidity. In the hydraulic fracturing stage, $CO_2$ can reduce the fracture pressure of the rock and contributes to the formation of a more complex fracture network. During the soak stage, $CO_2$ further diffuses and dissolves in water to form a weak acid environment, which can improve reservoir permeability and wettability, promote the conversion of oil-wetted surfaces to water-wetted surfaces, and contribute to the dissociation of residual oil. More importantly, $CO_2$ can supplement stratum energy and improve elastic displacement efficiency throughout the production stage. This helps to slow down the decay rate of stratum pressure and increase crude oil production. In 2019, the Xinjiang oilfield in China selected well J43-H for $CO_2$ fracturing and displacement test. Compared with the adjacent wells A, B and C, the pressure retention degree of well J43-H is 20% higher than those at the same fracturing fluid rejection rate, showing a promising ability to increase yield [87]. Although $CO_2$ fracturing and displacement technology have achieved the initial results, it is currently difficult to promote it in the Lucaogou Formation due to the insufficiency of the on-site workload. Future studies can focus on the $CO_2$ production enhancement mechanism based on indoor small-scale core tests and shale oil adsorption and desorption experiments, exploring the impact of $CO_2$ on the micro-occurrence of shale oil. And the program for $CO_2$ fracturing and displacement technology that suits the actual conditions of the work area should be developed. Basing on this foundation, field experiments can be encouraged to further validate and optimize $CO_2$ fracturing and displacement technology.

## Supporting information

**S1 Appendix. Minimal anonymous data, including two-dimensional nuclear magnetic resonance (2D NMR) data of rock samples, group composition, infrared spectroscopy, and chromatography data of crude oil, as well as daily oil production and oil pressure data.** (RAR)

## Acknowledgments

We would like to express our sincere gratitude to all those who contributed to the completion of this research. Our appreciation goes to Professor Lou for his guidance and support throughout the project. We also extend our thanks to the entire research team for their dedication and hard work. We are grateful to Hebei Scolike Petroleum Technology Co., Ltd for providing the necessary resources and facilities that made this research possible.

## Author Contributions

**Conceptualization:** Rong Zhu.

**Data curation:** Jiasi Li.

**Funding acquisition:** Zhanghua Lou.

**Investigation:** Jiasi Li.

**Methodology:** Jiasi Li.

**Project administration:** Zhanghua Lou.

**Software:** Jiasi Li.

**Supervision:** Aimin Jin, Rong Zhu, Zhanghua Lou.

**Writing – original draft:** Jiasi Li.

**Writing – review & editing:** Jiasi Li, Aimin Jin.

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
