## [Decision Letter · Decision Letter 0]

12 Sep 2023

PONE-D-23-18058Micro-occurrence characteristics and development response in continental shale oil from Lucaogou Formation in the Jimsar Sag, Junggar Basin, NW ChinaPLOS ONE

Dear Dr. Jin,

Thank you for submitting your manuscript to PLOS ONE. After careful consideration, we feel that it has merit but does not fully meet PLOS ONE’s publication criteria as it currently stands. Therefore, we invite you to submit a revised version of the manuscript that addresses the points raised during the review process.

We look forward to receiving your revised manuscript.

Kind regards,

Omeid Rahmani

Academic Editor

PLOS ONE

Journal Requirements:

2. We note that this submission includes NMR spectroscopy data. We would recommend that you include the following information in your methods section or as Supporting Information files:

1) The make/source of the NMR instrument used in your study, as well as the magnetic field strength. For each individual experiment, please also list: the nucleus being measured; the sample concentration; the solvent in which the sample is dissolved and if solvent signal suppression was used; the reference standard and the temperature.

2) A list of the chemical shifts for all compounds characterised by NMR spectroscopy, specifying, where relevant: the chemical shift (δ), the multiplicity and the coupling constants (in Hz), for the appropriate nuclei used for assignment.

3)The full integrated NMR spectrum, clearly labelled with the compound name and chemical structure.

We also strongly encourage authors to provide primary NMR data files, in particular for new compounds which have not been characterised in the existing literature. Authors should provide the acquisition data, FID files and processing parameters for each experiment, clearly labelled with the compound name and identifier, as well as a structure file for each provided dataset. 

See our list of recommended repositories here: https://journals.plos.org/plosone/s/recommended-repositories

"This research was financially supported by the National Science and Technology major projects (No. 2011ZX05002-006-003HZ)"

7. We note that Figure 1 in your submission contain map images which may be copyrighted. All PLOS content is published under the Creative Commons Attribution License (CC BY 4.0), which means that the manuscript, images, and Supporting Information files will be freely available online, and any third party is permitted to access, download, copy, distribute, and use these materials in any way, even commercially, with proper attribution. For these reasons, we cannot publish previously copyrighted maps or satellite images created using proprietary data, such as Google software (Google Maps, Street View, and Earth). For more information, see our copyright guidelines: http://journals.plos.org/plosone/s/licenses-and-copyright.

(1) You may seek permission from the original copyright holder of Figure 1 to publish the content specifically under the CC BY 4.0 license.  

8. We note that Figure 9 in your submission contain copyrighted images. All PLOS content is published under the Creative Commons Attribution License (CC BY 4.0), which means that the manuscript, images, and Supporting Information files will be freely available online, and any third party is permitted to access, download, copy, distribute, and use these materials in any way, even commercially, with proper attribution. For more information, see our copyright guidelines: http://journals.plos.org/plosone/s/licenses-and-copyright.

We require you to either (1) present written permission from the copyright holder to publish these figures specifically under the CC BY 4.0 license, or (2) remove the figures from your submission

(1) You may seek permission from the original copyright holder of Figure 9 to publish the content specifically under the CC BY 4.0 license. 

(2) If you are unable to obtain permission from the original copyright holder to publish these figures under the CC BY 4.0 license or if the copyright holder’s requirements are incompatible with the CC BY 4.0 license, please either i) remove the figure or ii) supply a replacement figure that complies with the CC BY 4.0 license. Please check copyright information on all replacement figures and update the figure caption with source information. 

If applicable, please specify in the figure caption text when a figure is similar but not identical to the original image and is therefore for illustrative purposes only.

Reviewers' comments:

Reviewer's Responses to Questions

**Comments to the Author**

1. Is the manuscript technically sound, and do the data support the conclusions?

Reviewer #1: Partly

2. Has the statistical analysis been performed appropriately and rigorously? 

Reviewer #1: N/A

3. Have the authors made all data underlying the findings in their manuscript fully available?

Reviewer #1: Yes

4. Is the manuscript presented in an intelligible fashion and written in standard English?

Reviewer #1: No

5. Review Comments to the Author

Reviewer #1: The method adopted by the authors is generally feasible. However, limited samples and poor article structure lead to reading difficulties. The following comments and suggestions may be useful to improve the quality of the article:

1) As for the title, "development response" is rarely discussed in this paper, and it is suggested to change it to "charging mechanism" or “accumulation mechanism”. In addition, the contents of section 4.6 do not belong to discussion, and are more suitable as prospects in the conclusion.

2) The abstract does not effectively summarize the manuscript. The abstract does not make clear what problems exist and what the significance of the study is. Suggest rewriting. There is no direct correlation between this research and the oil development raised by authors.

3) As for the introduction, three questions need to be answered clearly in the introduction: what research question(s) do the authors address? Do they make a good argument for why a question is important? What methods do the authors use to answer the question? The authors only raised one existing question in the introduction, that is, the extraction standard is not uniform. It is suggested to rewrite the introduction.

4) The author provides a lot of figures with simple figure name. The information and conclusions of these figures should be described in detail in figure caption to facilitate readers' better understanding. Some of the information in the figure is not even found in the text; for example, the formation abbreviation and oil saturation in Figure 1 are not stated.

5) The XRD results and figures in Chapter 3 should be transferred to Chapter 4. The authors are advised to add the duration range and temperature of extraction. It is recommended to supplement the instruments, experimental parameters and laboratories for all experiments.

6）It is suggested that the authors divide the results and discussion in Chapter 4 into two chapters. Given that the rock types of the Lucaogou Formation are very complex and diverse, the number of samples collected by the authors is slightly insufficient (only two), which may lead readers to question the statistical significance of the results. In addition, the authors do not discuss how representative the sample is to the stratum, and whether only a few samples can be representative of the whole stratum.

7) The historical process of oil charging is not clearly described. The source of the buried history data needs to be explained. There should be an obvious uplift denudation of K2-E, leading to large-scale denudation of the upper part of Lusaogou Formation in Jimsar Sag.

6. PLOS authors have the option to publish the peer review history of their article (what does this mean?). If published, this will include your full peer review and any attached files.

Reviewer #1: No

---

## [Author Response · Author response to Decision Letter 0]

21 Nov 2023

1) As for the title, "development response" is rarely discussed in this paper, and it is suggested to change it to "charging mechanism" or “accumulation mechanism”. In addition, the contents of section 4.6 do not belong to discussion, and are more suitable as prospects in the conclusion.

Response: Thank you. In response to your valuable suggestion, we have made the recommended modifications. Firstly, we have revised the title to better align with the content of the paper. "Development response" has been changed to "charging mechanism" to accurately reflect the focus of the study. Additionally, we have addressed the concern regarding the placement of content in section 4.6. We have relocated this section from the discussion to the conclusion, where it is now presented as a new chapter labeled "Prospects."

2) The abstract does not effectively summarize the manuscript. The abstract does not make clear what problems exist and what the significance of the study is. Suggest rewriting. There is no direct correlation between this research and the oil development raised by authors.

Response: We appreciate the feedback provided. In response to your comments, we have thoroughly reviewed and amended the abstract to more effectively summarize the manuscript. This revision places a stronger emphasis on the issues addressed in our research and the significance of our findings. While the study may not directly guide shale oil development, it does offer valuable insights into the micro-occurrence characteristics of shale oil, shedding light on its microscopic attributes, mobility, and potential applications in refining extraction processes. We trust that these revisions will better align the abstract with the core aspects of our study.

3) As for the introduction, three questions need to be answered clearly in the introduction: what research question(s) do the authors address? Do they make a good argument for why a question is important? What methods do the authors use to answer the question? The authors only raised one existing question in the introduction, that is, the extraction standard is not uniform. It is suggested to rewrite the introduction.

Response: Thank you for your valuable guidance. We have carefully restructured the introduction in response to your recommendations. The revised introduction now explicitly addresses the research questions we are investigating, provides a compelling argument for the significance of these questions, and thoroughly explains the methods employed to address these inquiries. We believe this restructured introduction successfully aligns with your feedback, and we appreciate your assistance in enhancing the clarity of our manuscript.

4) The author provides a lot of figures with simple figure name. The information and conclusions of these figures should be described in detail in figure caption to facilitate readers' better understanding. Some of the information in the figure is not even found in the text; for example, the formation abbreviation and oil saturation in Figure 1 are not stated.

Response: Thank you very much for your suggestion. We have thoroughly reviewed all figure captions and made modifications and additions to each of them, providing more detailed information related to the figures. We believe that these revisions will significantly enhance readers' understanding of the figures.

5) The XRD results and figures in Chapter 3 should be transferred to Chapter 4. The authors are advised to add the duration range and temperature of extraction. It is recommended to supplement the instruments, experimental parameters and laboratories for all experiments.

Response: Firstly, we acknowledge the suggestion to move the XRD results from Chapter 3 to Chapter 4. However, upon careful consideration, we have decided to remove the XRD results from the manuscript. After a comprehensive review, we found that the XRD results played a limited role in the paper and did not significantly contribute to the overall content.

Secondly, we have addressed the recommendation by adding details about the experimental instruments, parameters, and laboratory information for all experiments. This addition should provide readers with a clearer understanding of the experimental setup and conditions.

6）It is suggested that the authors divide the results and discussion in Chapter 4 into two chapters. Given that the rock types of the Lucaogou Formation are very complex and diverse, the number of samples collected by the authors is slightly insufficient (only two), which may lead readers to question the statistical significance of the results. In addition, the authors do not discuss how representative the sample is to the stratum, and whether only a few samples can be representative of the whole stratum.

Response: Thank you for your suggestion. After careful consideration, we have decided to keep the results and discussion in Chapter 4 together, as this structure aligns better with the overall flow of the paper.

Regarding the issue of sample quantity and representativeness, we have provided additional explanations in the "Samples and Experiments" section. While we acknowledge that the number of samples is limited, they do represent the primary oil-bearing layers and lithologies of the Lucaogou Formation, making them reasonably representative. Through the study of these samples, we aim to contribute to a comprehensive understanding of the micro-occurrence characteristics of shale oil in the Lucaogou Formation.

7) The historical process of oil charging is not clearly described. The source of the buried history data needs to be explained. There should be an obvious uplift denudation of K2-E, leading to large-scale denudation of the upper part of Lusaogou Formation in Jimsar Sag.

Response: Thank you for your suggestion. We have revisited the description of the historical process of oil charging and provided an explanation regarding the source of buried history data.

In reference to your comment about the "obvious uplift denudation of K2-E," we may need further clarification on this point as it wasn't entirely clear. However, we believe that this should not significantly impact the overall micro-filling process of shale oil within the context of the charging history. If you have any further questions or clarifications, please feel free to let us know, and we will address them accordingly.

---

## [Editor Report · Decision Letter 1]

27 Dec 2023

Micro-occurrence characteristics and charging mechanism in continental shale oil from Lucaogou Formation in the Jimsar Sag, Junggar Basin, NW China

PONE-D-23-18058R1

Dear Aimin Jin,

We’re pleased to inform you that your manuscript has been judged scientifically suitable for publication and will be formally accepted for publication once it meets all outstanding technical requirements.

Kind regards,

Omeid Rahmani

Academic Editor

PLOS ONE

Additional Editor Comments (optional):

Put the XRD results as supplementary data.
---

## [Editor Report · Acceptance letter]

26 Jan 2024

PONE-D-23-18058R1 

PLOS ONE

Dear Dr. Jin, 

I'm pleased to inform you that your manuscript has been deemed suitable for publication in PLOS ONE. Congratulations! Your manuscript is now being handed over to our production team.

Kind regards, 

on behalf of

Dr. Omeid Rahmani 

Academic Editor

PLOS ONE